EMBO
Molecular Medicine

# Targeting HPK1 inhibits neutrophil responses to mitigate post-stroke lung and cerebral injuries

Tingting Zhang [ID][1,2,5], Ying Sun[1,2,5], Jing Xia[1,2], Hongye Fan[1,2], Dingfang Shi[1,2], Qian Wu [ID][3], Ming Huang [ID][4✉] & Xiao-Yu Hou [ID][1,2✉]

## Abstract

**Circulating neutrophils are responsible for poor neurological outcomes and have been implicated in respiratory morbidity after acute ischemic stroke (AIS). However, the molecular mechanisms regulating neutrophil responses and their pathological relevance in post-stroke complications remain unclear. In this study, we investigated the involvement of hematopoietic progenitor kinase 1 (HPK1) in neutrophil responses and mobilization, as well as subsequent lung and cerebral injuries following AIS. We found that lipopolysaccharide treatment triggered neutrophil activation in an HPK1-dependent manner. HPK1 enhanced intrinsic NF-κB/STAT3/p38-MAPK pathways and gasdermin D cleavage, leading to neutrophil hyperactivation. Following AIS, HPK1 promoted the mobilization of CXCR2^high bone marrow neutrophils. HPK1 loss inhibited peripheral neutrophil hyperactivation, neutrophil infiltration, and aggregation of neutrophil extracellular traps, progressively alleviating systemic inflammation and impairments in mouse pulmonary and neurological functions. Furthermore, HPK1 pharmacological inhibition attenuated post-stroke pulmonary and neurological impairments in mice. Our findings revealed that HPK1 upregulates neutrophil mobilization and various responses, promoting post-stroke systemic inflammation and tissue injury. This study highlights HPK1 as a therapeutic target for improving pulmonary and neurological functions after AIS.**

**Keywords** Acute Ischemic Stroke; Gasdermin D; Neutrophil Extracellular Traps; Neutrophil Mobilization; Systemic Inflammation
**Subject Categories** Cardiovascular System; Neuroscience; Vascular Biology & Angiogenesis

## Introduction

Acute ischemic stroke (AIS) is the leading cause of adult disability; moreover, its increasing incidence, particularly in young adults, poses a public health challenge (Katan and Luft, 2018; Yahya et al, 2020). In addition to ischemic cerebral damage, post-stroke impairments in peripheral organs, particularly vulnerable lungs, lead to worse outcomes and a high mortality risk in AIS survivors (Bai et al, 2018; Robba et al, 2020; Wang et al, 2022). AIS triggers systemic inflammatory responses associated with the development of post-stroke neurological and pulmonary complications (Samary et al, 2018; Endres et al, 2022; Simats and Liesz, 2022). Understanding the cellular and molecular mechanisms mediating the post-stroke inflammatory state and systemic injury will help identify potential molecular targets for better outcomes.

Polymorphonuclear neutrophils, the most abundant circulating leukocytes, respond rapidly and persistently to ischemic insults (Jickling et al, 2015; Wanrooy et al, 2021). In patients with AIS, peripheral blood neutrophil counts increase during the hyperacute and acute phases, lasting from the first few hours to over 7 d after stroke onset (Ross et al, 2007; Jickling et al, 2015). Rapid increases in circulating leukocyte number or the neutrophil-to-lymphocyte ratio (NLR) predict poor neurological outcomes and mortality in patients undergoing reperfusion therapy for AIS (Tian et al, 2018; Bi et al, 2021; Sharma et al, 2021). NLR also predicts post-stroke pneumonia risk and severity, with most sterile pneumonia cases occurring within 7 d of AIS onset (Nam et al, 2018a; Nam et al, 2018b; Li et al, 2023). Upon infection or sterile inflammation, neutrophils are mobilized into the blood, where activated neutrophils release proteases, reactive oxygen species (ROS), and inflammatory mediators (Kolaczkowska and Kubes, 2013; Liew and Kubes, 2019). Overactivated neutrophils release neutrophil extracellular traps (NETs) that are web-like structures formed by neutrophil nuclei and granzymes through the NETosis pathway. NETs are observed in the ischemic hemisphere of patients with AIS, and elevated plasma NETs affect neurofunctional recovery in mice (Denorme et al, 2022; Jiang et al, 2024). These studies have implicated acute-phase-mobilized neutrophils in neurological and pulmonary dysfunction after AIS. However, the regulation of neutrophil responses and mobilization following AIS and their pathological relevance in post-stroke systemic injury remain unclear.

Hematopoietic progenitor kinase 1 (HPK1), also known as mitogen-activated protein kinase kinase kinase kinase 1 (MAP4K1),

[1]State Key Laboratory of Natural Medicines, China Pharmaceutical University, Nanjing, Jiangsu 211198, China. [2]School of Life Science and Technology, China Pharmaceutical University, Nanjing, Jiangsu 211198, China. [3]The Department of Neurology, The First Affiliated Hospital of Nanjing Medical University, Nanjing, Jiangsu 210029, China. [4]Research Center for Biochemistry and Molecular Biology, Xuzhou Medical University, Xuzhou, Jiangsu 221004, China. [5]These authors contributed equally: Tingting Zhang, Ying Sun. ✉E-mail: huangming88@xzhmu.edu.cn; xyhou@cpu.edu.cn

is predominantly expressed in hematopoietic cells, including myeloid cells and lymphocytes, and lymphoid organs/tissues (Hu et al, 1996; Kiefer et al, 1996). HPK1 promotes CXC chemokine ligand 1-induced neutrophil adhesion under flow conditions, probably through phosphorylating adaptor protein HPK1-interacting protein of 55 kDa (Schymeinsky et al, 2009; Jakob et al, 2013; Chuang et al, 2016). Moreover, HPK1 positively regulates neutrophil adhesion and recruitment following tumor necrosis factor (TNF)-α-induced acute local inflammation in mouse cremaster muscle (Jakob et al, 2013). Recently, the roles of T cell- and cancer cell-intrinsic HPK1 in tumor immunosuppression have been identified (Hernandez et al, 2018; Si et al, 2020; Sun et al, 2023). However, the functional impact of HPK1 on innate immunity and its contribution to pathological processes following acute brain disorders remain unknown.

This study aimed to investigate whether and how HPK1 is involved in neutrophil inflammatory responses and NET formation, either in vitro following lipopolysaccharide (LPS) treatment or in mice with ischemic stroke followed by reperfusion. Furthermore, we investigated the involvement of HPK1-regulated neutrophil mobilization and infiltration in systemic inflammation and functional deficits in lung and brain tissues following AIS. This study suggests a novel therapeutic strategy for stroke-associated acute lung and cerebral injuries.

# Results

## LPS upregulates HPK1 and induces primary neutrophil hyperactivity

First, we investigated the effects of HPK1 on neutrophil activity in response to LPS stimulation. Primary neutrophils were isolated from the bone marrow of adult HPK1 knockout (KO, *Map4k1⁻/⁻*) mice or their wild-type (WT, *Map4k1⁺/⁺*) littermate controls and stimulated with LPS (Fig. 1A). Quantitative polymerase chain reaction (qPCR) and immunoblot confirmed the loss of HPK1 in neutrophils from KO mice (Fig. 1B,C). LPS (10, 100, and 1000 ng/mL) stimulation upregulated proinflammatory factors in primary neutrophils from WT mice, including *Tnf*, *Il6*, *Il1b*, and *Nos2* mRNA levels (Fig. 1D), whereas HPK1 levels increased in a sustained manner (Fig. EV1A,B). In contrast, HPK1 loss reduced proinflammatory responses (Fig. 1E–H) and ROS production (Fig. 1I,J) following LPS (100 ng/mL) treatment. As NET formation is a neutrophil hyperactivation hallmark (Papayannopoulos, 2018), the nucleic acid dye SYTOX Green was used to mark neutrophils undergoing NETosis and extracellular NETs. LPS (10 μg/mL) stimulated HPK1 upregulation (Fig. EV1C,D) and NET formation (Fig. 1K,L), whereas NET formation was markedly reduced in HPK1-KO neutrophils compared with that in WT cells (Fig. 1K,L). Under physiological conditions, HPK1-KO neutrophils exhibited levels of proinflammatory factor transcripts (Fig. 1E–H), ROS (Fig. 1I,J), and NETs (Fig. 1K,L) comparable to those of WT controls. These data suggest that neutrophil HPK1 upregulation mediates inflammatory and oxidative responses and NET formation.

## Neutrophil HPK1 induces inflammatory and gasdermin D (GSDMD) signal pathways after LPS stimulation

We determined the molecular mechanisms by which HPK1 mediates proinflammatory responses and NET formation in

neutrophils. Nuclear factor-κB (NF-κB), signal transducer and activator of transcription 3 (STAT3), and p38-MAPK are known proinflammatory signaling pathways. LPS (100 ng/mL) stimulation increased NF-κB (p-NF-κB) and IκBα (p-IκBα) phosphorylation but not their total protein levels in primary neutrophils; these increases were abolished after HPK1 KO (Fig. 2A–C). Consistently, STAT3 (p-STAT3) and p38-MAPK (p-p38) phosphorylation significantly increased in WT neutrophils after LPS stimulation, whereas HPK1 loss reduced these increases (Fig. 2A,D,E).

GSDMD, an executor of pyroptosis-like cell death, is required for pyroptosis-independent release of proinflammatory factors in neutrophils (Heilig et al, 2018; Broz et al, 2020; Karmakar et al, 2020). GSDMD activation (cleavage) is also linked with NETosis (Sollberger et al, 2018). Nigericin, an activator of NLRP3 inflammasome, combined with LPS (500 ng/mL) priming (LN) upregulated the levels of cleaved and active GSDMD forms (N-GSDMD) and N-GSDMD-to-total GSDMD ratio (N-GSDMD/GSDMD) in WT neutrophils (Fig. 2F–H). However, HPK1 KO reduced N-GSDMD levels and N-GSDMD/GSDMD ratio compared with the WT groups (Fig. 2F–H).

These results suggest that neutrophil-intrinsic HPK1 aggravates inflammatory responses and NETosis by promoting NF-κB/STAT3/p38-MAPK and GSDMD pathway activation.

## HPK1 is responsible for circulating neutrophil hyperactivation in mice after ischemic stroke

Next, we determined whether circulating neutrophil HPK1 is involved in peripheral inflammation and NET formation in mice following ischemic stroke. As shown in Fig. 3, HPK1 levels were upregulated in neutrophils isolated from the peripheral blood of WT mice (Fig. 3A,B) and neutrophils accumulated rapidly in the blood of WT mice (Fig. 3C,D) after middle cerebral artery occlusion (60 min) followed by reperfusion (MCAO/R) for 6 h. The number and frequency of peripheral neutrophils increased persistently, whereas HPK1 KO induced their decreases in mice subjected to MCAO/R for 24 h but not in sham-operated mice (Fig. 3E,F). The levels of plasma proinflammatory cytokines, including TNF-α, interleukin (IL)-6, and IL-1β, consistently reduced in HPK1-KO mice compared with that in WT mice after MCAO/R (Fig. 3G–I). Similarly, ROS production in peripheral blood neutrophils was diminished in HPK1-KO mice after MCAO/R (Fig. 3J). To determine the contribution of HPK1 to post-stroke formation of circulating NETs, we isolated peripheral blood neutrophils after MCAO/R for 24 h. Fluorescence imaging showed that peripheral blood neutrophils from post-stroke WT mice had a high NET ratio following LPS (5 μg/mL) treatment, whereas such NET ratio was effectively inhibited in HPK1-KO neutrophils (Fig. 3K,L). Therefore, neutrophil HPK1 contributes to systemic inflammatory and oxidative responses and NET formation in mice following ischemic stroke.

## Neutrophil HPK1 facilitates mouse bone marrow neutrophil mobilization by upregulating CXC chemokine receptor 2 (CXCR2) following ischemic stroke

Most neutrophils reside in the bone marrow under normal physiological conditions and are quickly mobilized into the blood upon injury (Borregaard, 2010). To investigate whether HPK1 is

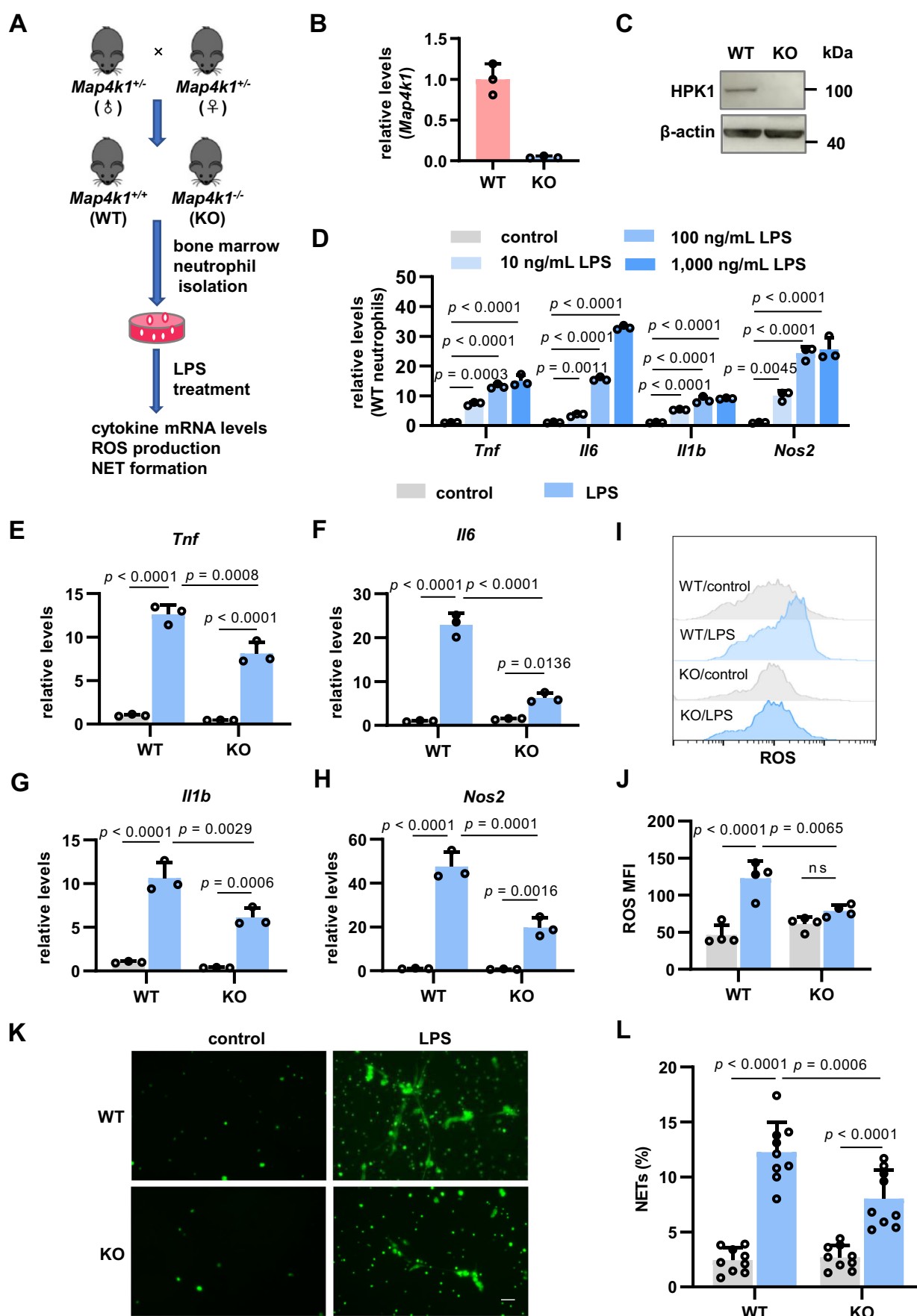

◄ **Figure 1.    HPK1 mediates the hyperactivation of primary neutrophils after LPS treatment.**

(A) Experimental scheme for analysis of primary bone marrow neutrophil responses. (B, C) qPCR (B) and immunoblot (C) analysis of HPK1 expression in primary neutrophils from HPK1-KO ($Map4k1^{-/-}$) mice and its littermate WT controls ($Map4k1^{+/+}$). $N = 3$ per group. (D) qPCR analysis of $Tnf$, $Il6$, $Il1b$, and $Nos2$ in neutrophils from WT mice treated with different LPS concentrations. $N = 3$ per group, three-time repeats. (E–H) qPCR analysis of $Tnf$ (E), $Il6$ (F), $Il1b$ (G), and $Nos2$ (H) in neutrophils from WT and KO mice treated with LPS (100 ng/mL). $N = 3$ per group, three-time repeats. (I, J) Representative flow cytometry analysis (I) and mean fluorescence intensity (MFI) values (J) of ROS production in neutrophils from WT and KO mice treated with LPS (100 ng/mL). $N = 4$ per group, three-time repeats. (K, L) Representative fluorescence imaging (K) and quantification (L) of NET formation in neutrophils from WT and KO mice. Neutrophils were treated with LPS (10 μg/mL) for 2.5 h. NETs were visualized using SYTOX Green. Scale bar = 50 μm. $N = 9$ per group. Bar graphs represent the mean ± SD. ns, no significance. One-way ANOVA in (D), two-way ANOVA in (E–H), (J), and (L). Source data are available online for this figure.

associated with neutrophil mobilization following ischemic stroke, we isolated and analyzed the neutrophil population of mouse bone marrow. As shown in Fig. 4, the number and frequency of bone marrow neutrophils were much lower in WT mice subjected to MCAO/R for 6 h (Fig. 4A,B) and 24 h (Fig. 4C,D) than in the respective sham groups. HPK1 KO did not alter the neutrophil population of sham groups but reversed its post-stroke decline in the bone marrow (Fig. 4C,D), suggesting that HPK1 promotes the mobilization of bone marrow neutrophils after ischemic stroke.

CXCR2 signaling is required for the mobilization of bone marrow neutrophils, while CXCR4 signaling accounts for neutrophil retention in bone marrow (Eash et al, 2010). We found that membrane CXCR2 levels were elevated and membrane CXCR4 levels did not change in bone marrow neutrophils after MCAO/R for 24 h, while HPK1 loss reversed this increase in membrane CXCR2 levels (Fig. 4E,F). Total CXCR2 levels were consistently upregulated in bone marrow neutrophils after MCAO/R, whereas HPK1 loss reversed this increase (Fig. 4G,H).

These findings suggest that HPK1 promotes the mobilization of bone marrow neutrophils by upregulating CXCR2 signaling after ischemic stroke.

## HPK1 mediates neutrophil infiltration and hyperactivation in mouse lung and brain tissues after ischemic stroke

Furthermore, we investigated whether HPK1 contributes to local inflammation and tissue injury within the acute phase after AIS (Fig. 5A). Neutrophil accumulation into the lungs was greater in WT mice subjected to MCAO/R for 24 h than in the sham groups, whereas HPK1 loss reduced this accumulation (Fig. 5B,C). In contrast, although MCAO/R also upregulated the number and frequency of neutrophils in spleen tissues, HPK1 loss did not influence the post-stroke spleen neutrophil population (Fig. EV2A,B). Proinflammatory cytokines in the lungs of HPK1-KO mice were downregulated compared with those of WT mice following MCAO/R for 24 h (Fig. 5D–F). In addition, the levels of citrullinated histone H3 (CitH3), a specific marker of NETosis, were much lower in the lungs of male and female HPK1-KO mice than in WT groups after MCAO/R for 72 h (Fig. 5G,H). Lung tissues showed alveolar septal thickening and leukocyte infiltration in WT mice after MCAO/R for 72 h (Fig. 5I,J). Total protein concentration in the bronchoalveolar lavage fluid (BALF), a measure of alveolar-capillary barrier disruption, was consistently upregulated in WT mice after MCAO/R (Fig. 5K). However, HPK1 loss alleviated these lung injuries (Fig. 5I–K). The sham operation did not induce marked NET formation (Fig. EV3A,B) and lung injury in WT and HPK1-KO mice (Fig. EV3C–E).

We also determined the extent of brain inflammation and injury following ischemic stroke in WT and HPK1-KO mice (Fig. 6A). Neutrophil number and frequency in the MCAO/R hemisphere were greater than those in the control hemisphere after MCAO/R for 24 h, whereas HPK1 KO suppressed neutrophil infiltration into the MCAO/R hemisphere (Fig. 6B,C). Compared with WT controls, HPK1 KO led to decreased TNF-α, IL-6, and IL-1β levels in the ischemic hemisphere after MCAO/R for 24 h (Fig. 6D–F). CitH3 immunofluorescence (IF) analysis showed that MCAO/R for 72 h induced NETosis in the MCAO/R brain tissues but not its control hemisphere (Fig. 6G–I); HPK1 loss reduced NETosis in the mouse brain (Fig. 6G,I). Consistently, 2,3,5-triphenyitetrazolium chloride (TTC) staining combined with modified neurological severity score (mNSS) evaluation showed that HPK1 loss attenuated cerebral infarct size (Fig. 6J,K) and neurological severity (Fig. 6L) in mice after MCAO/R for 72 h. We used both sexes and found no significant effect of sex on post-stroke outcomes measured (Figs. 5G–K and 6J–L).

These results suggest that the HPK1 mediates neutrophil infiltration, inflammatory responses, and neutrophil NETosis, at least in part, thereby aggravating acute lung injury and neurological deficits following ischemic stroke.

## Therapy with HPK1 inhibitor ameliorates neutrophil infiltration and lung injury after ischemic stroke

To further verify that HPK1 mediates neutrophil responses, we pretreated primary neutrophils with a small-molecule HPK1 inhibitor (i-HPK1) followed by LPS stimulation. As shown in Fig. EV4A–C, i-HPK1 decreased LPS-induced upregulation of $Tnf$, $Il6$, $Il1b$, $Nos2$, and $Ptgs2$ mRNA levels but did not affect the upregulated anti-inflammatory cytokines, including $Socs3$ and $Il10$ mRNA levels. Flow cytometry analysis showed that i-HPK1 pretreatment reduced LPS-stimulated ROS production in neutrophils (Fig. EV4D). Pretreatment with i-HPK1 also reduced LPS-induced NET formation (Fig. EV4E,F). Mechanically, i-HPK1 inhibited p-NF-κB, p-IκBα, p-STAT3, and p-p38 (Fig. EV4G–K) as well as GSDMD cleavage levels (Fig. EV4L–N) in primary neutrophils.

To evaluate the therapeutic potential of i-HPK1 in ameliorating post-stroke lung injury, i-HPK1 (2 mg/kg) was administered 30 min after reperfusion following a severe ischemia (90 min of occlusion). Histopathological analysis showed that i-HPK1 therapy attenuated lung injury in mice following MCAO (90 min)/R for 48 h (Fig. 7A,B). Total protein levels in BALF were lower in i-HPK1-posttreated mice than in vehicle-treated mice (Fig. 7C). Moreover, i-HPK1 therapy inhibited neutrophil infiltration (Fig. 7D,E) and NETosis (Fig. 7F,G) in the lungs after MCAO (90 min)/R.

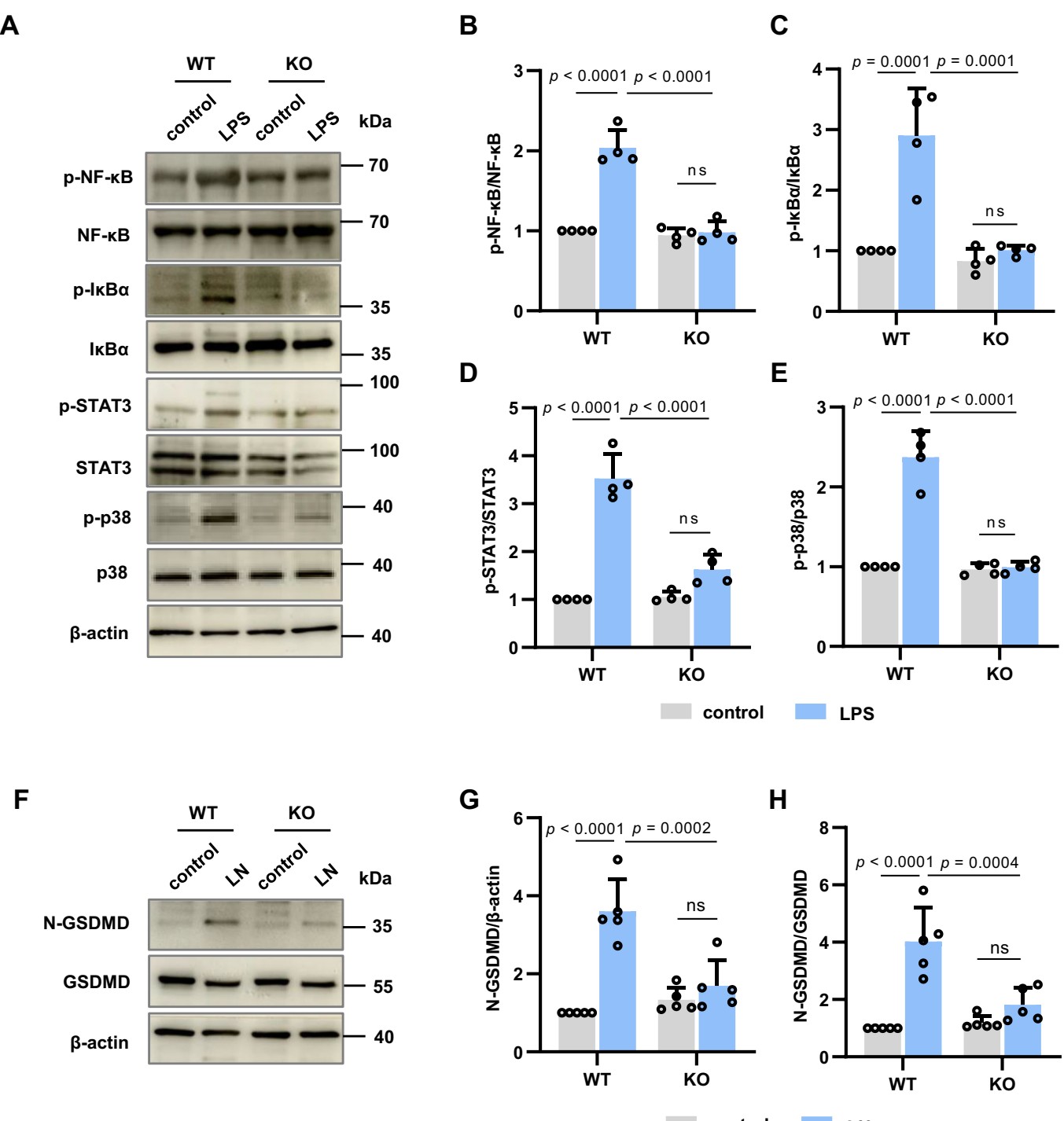

**Figure 2. HPK1 mediates proinflammatory NF-κB/STAT3/p38 pathways and GSDMD cleavage in primary neutrophils after LPS treatment.**

(A–E) Representative immunoblot analysis (A) and quantification of phosphorylation and total protein levels of NF-κB (B), IκBα (C), STAT3 (D), and p38 (E) in primary neutrophils from WT and KO mice. Primary neutrophils were treated with LPS (100 ng/mL) for 1 h in NF-κB, IκBα, and p38 phosphorylation analysis, or 3 h in STAT3 phosphorylation analysis. N = 4 per group. (F–H) Representative immunoblot analysis (F) and quantification of GSDMD cleavage (N-GSDMD) levels (G) and N-GSDMD/GSDMD ratio (H) in neutrophils from WT and KO mice treated with LPS (500 ng/mL) and nigericin (LN). N = 5 per group. Bar graphs represent the mean ± SD. ns, no significance. Two-way ANOVA in (B–E), (G), and (H). Source data are available online for this figure.

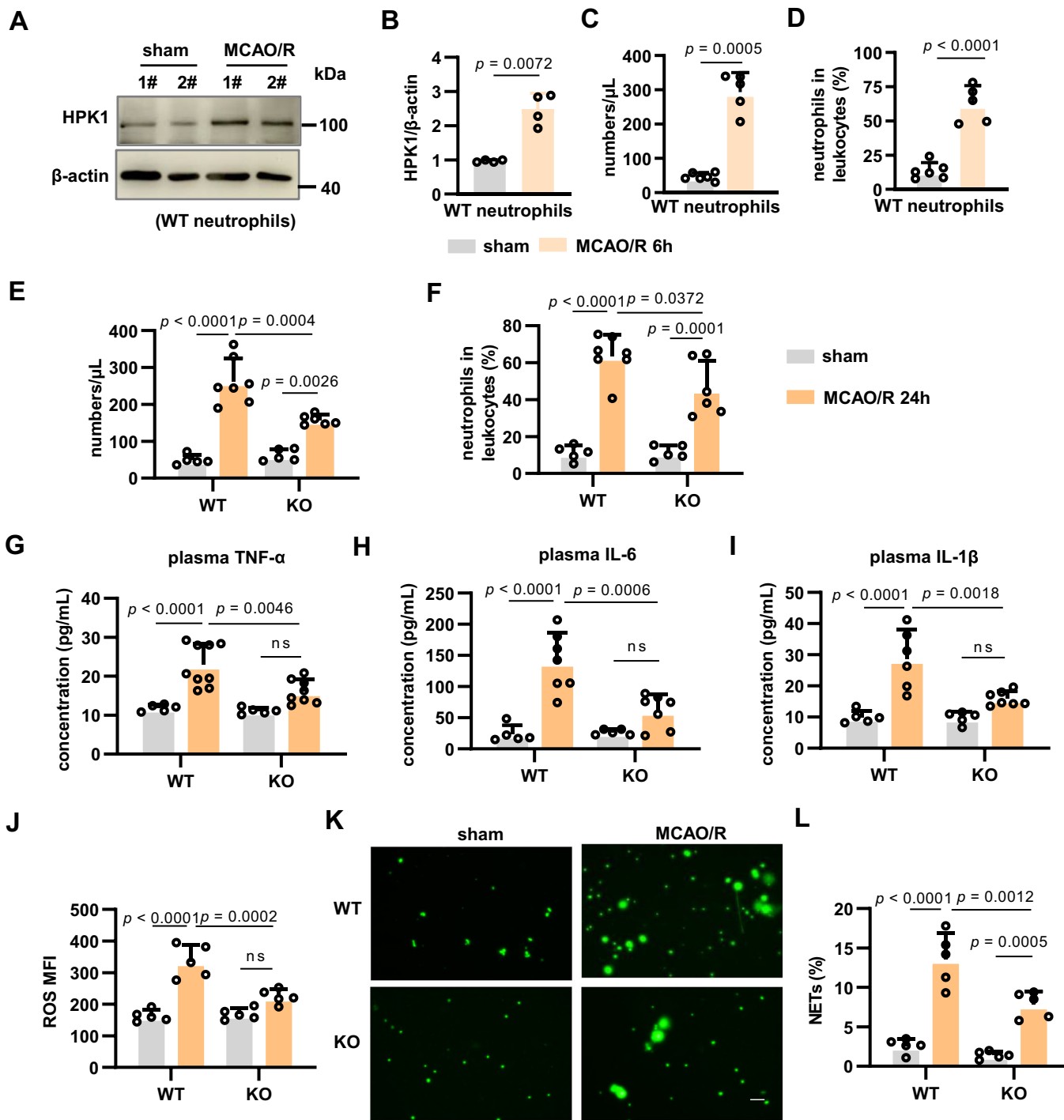

**Figure 3. HPK1 promotes peripheral inflammation and neutrophil responses in mice after ischemic stroke.**

(A, B) Representative HPK1 immunoblot (A) and quantification (B) in peripheral blood neutrophils from WT mice after sham or MCAO/R for 6 h. $N = 4$ mice per group. (C, D) Flow cytometry analysis of the number (C) and frequency (D) of peripheral blood neutrophils from WT mice after sham or MCAO/R for 6 h. $N = 5$–6 mice per group. (E, F) Flow cytometry analysis of the number (E) and frequency (F) of peripheral blood neutrophils from WT/KO mice after sham or MCAO/R for 24 h. $N = 5$–7 mice per group. (G–I) ELISA assay of plasma TNF-α (G), IL-6 (H), and IL-1β (I) levels in WT/KO mice after sham or MCAO/R for 24 h. $N = 5$–9 mice per group. (J) Flow cytometry analysis of ROS production (MFI) in peripheral blood neutrophils from WT/KO mice after sham or MCAO/R for 24 h. $N = 5$ mice per group. (K, L) Representative fluorescence imaging (K) and quantification (L) of NET formation in peripheral blood neutrophils. Isolated neutrophils were stimulated with LPS (5 μg/mL) for 2.5 h, and NETs were visualized using SYTOX Green. Scale bar = 50 μm. $N = 5$ mice per group. Bar graphs represent the mean ± SD. ns, no significance. Unpaired Student's t-test in (B–D), two-way ANOVA in (E–J) and (L). Source data are available online for this figure.

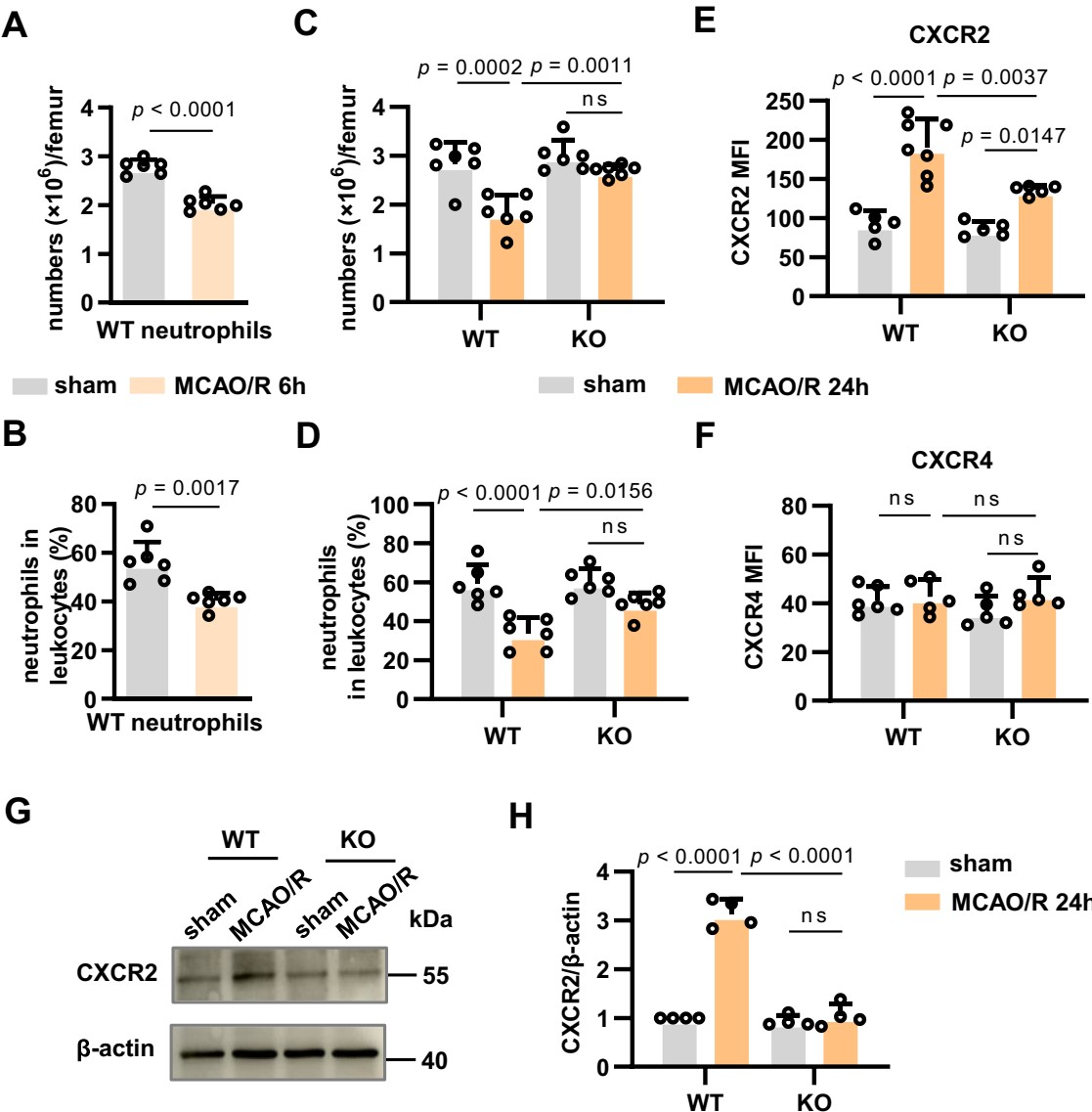

**Figure 4. HPK1 promotes bone marrow neutrophil mobilization by modulating CXCR2 expression in mice after ischemic stroke.**

(A, B) Flow cytometry analysis of the number (A) and frequency (B) of bone marrow neutrophil population in WT mice after sham or MCAO/R for 6 h. $N = 6$ mice per group. (C, D) Flow cytometry analysis of the number (C) and frequency (D) of bone marrow neutrophil population in WT/KO mice after sham or MCAO/R for 24 h. $N = 6$ mice per group. (E, F) Flow cytometry analysis of membrane CXCR2 (E) and membrane CXCR4 (F) levels (MFI) of bone marrow neutrophils in WT/KO mice after sham or MCAO/R for 24 h. $N = 5$-7 mice per group. (G, H) Representative CXCR2 immunoblot (G) and quantification (H) in bone marrow neutrophils from WT/KO mice after sham or MCAO/R for 24 h. $N = 4$ mice per group. Bar graphs represent the mean ± SD. ns, no significance. Unpaired Student's $t$-test in (A, B) and two-way ANOVA in (C–F) and (H). Source data are available online for this figure.

These results validate the molecular mechanisms whereby HPK1 mediates proinflammatory responses and NET formation in neutrophils and suggest that i-HPK1 treatment ameliorates acute lung injury after ischemic stroke.

## Therapy with HPK1 inhibitor ameliorates brain injury and neuroinflammation after ischemic stroke

Finally, we examined post-stroke neurological impairment and neuroinflammation following i-HPK1 posttreatment. We found that i-HPK1 therapy reduced cerebral infarct volume in TTC

staining (Fig. 8A,B) and neurological severity in the mNSS evaluation (Fig. 8C) after MCAO (90 min)/R for 48 h. Flow cytometry analysis showed that i-HPK1 treatment inhibited neutrophil infiltration at the early reperfusion stage (MCAO/R for 24 h) (Fig. 8D,E) but had no influence on bone marrow neutrophil release (Fig. EV5A,B). Glia-related neuroinflammatory responses are responsible for the pathological process after stroke (Dong et al, 2021; Shi et al, 2025). We found that CD86 levels in microglia were reduced in i-HPK1 treatment mice compared with that in the MCAO/R group (Fig. 8F). In addition, CitH3 IF analysis showed that i-HPK1 treatment reduced NET formation in the

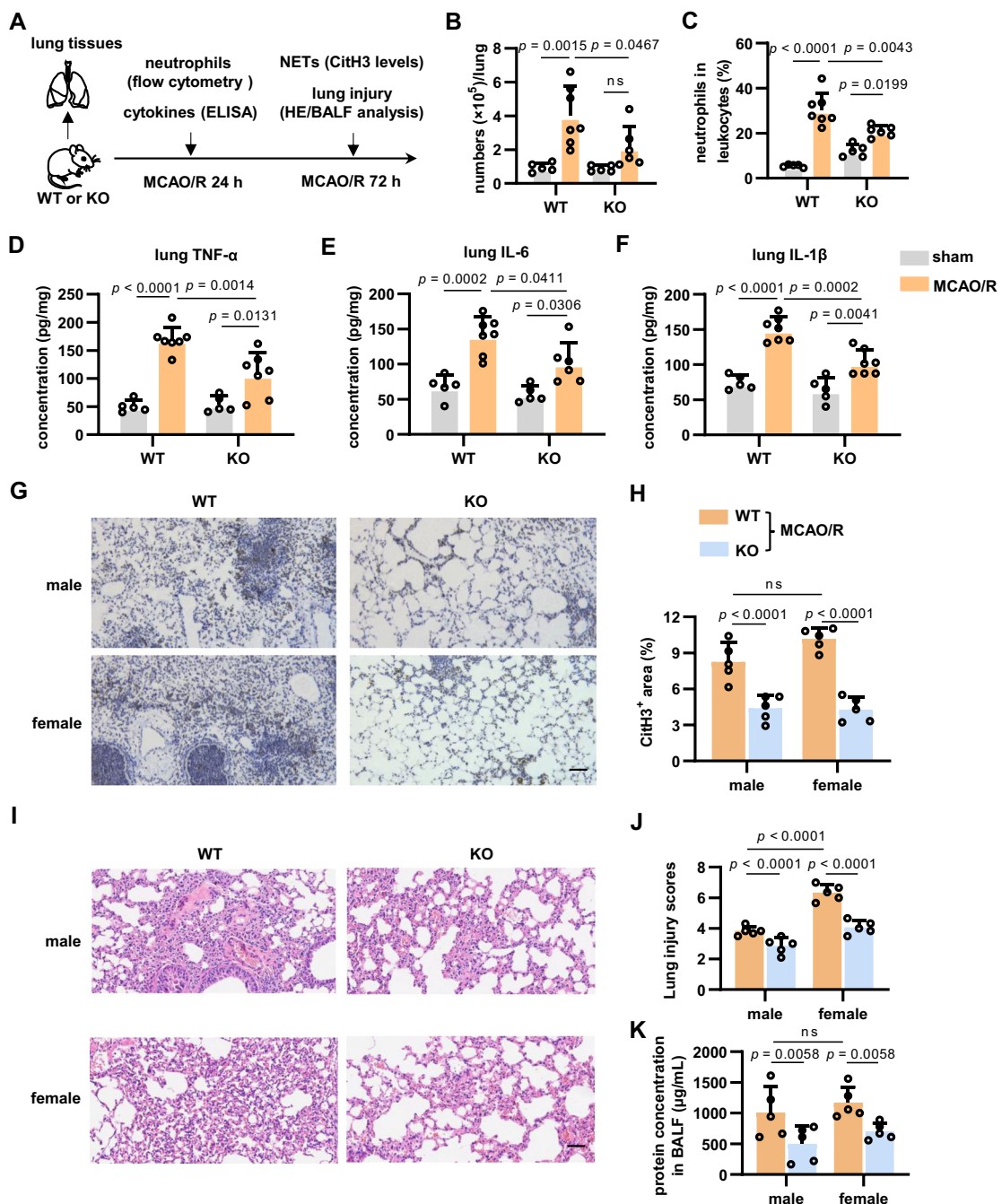

**Figure 5. HPK1 loss ameliorates lung inflammation and injury in mice after ischemic stroke.**

(A) Schematic showing the experimental procedures for assessing lung inflammation and injury in WT/KO mice after MCAO/R. (B, C) Flow cytometry analysis of number (B) and frequency (C) of infiltrated neutrophils in the lung tissues after sham or MCAO/R for 24 h. $N = 5–7$ mice per group. (D–F) ELISA assay of TNF-α (D), IL-6 (E), and IL-1β (F) levels in the lung tissues after sham or MCAO/R for 24 h. $N = 5–7$ mice per group. (G, H) Representative CitH3 immunohistochemical images (G) and quantification (H) in the lung sections from male and female mice after MCAO/R for 72 h. $N = 5$ mice per group. Scale bar = 50 μm. (I, J) Representative HE staining (I) and histopathological scores (J) of lung sections from male and female mice after MCAO/R for 72 h. Scale bar = 50 μm. $N = 5$ mice per group. (K) Protein concentration in BALF from male and female mice after MCAO/R for 72 h. $N = 5$ mice per group. Bar graphs represent the mean ± SD. ns, no significance. Two-way ANOVA in (B–F), (H), (J), and (K). Source data are available online for this figure.

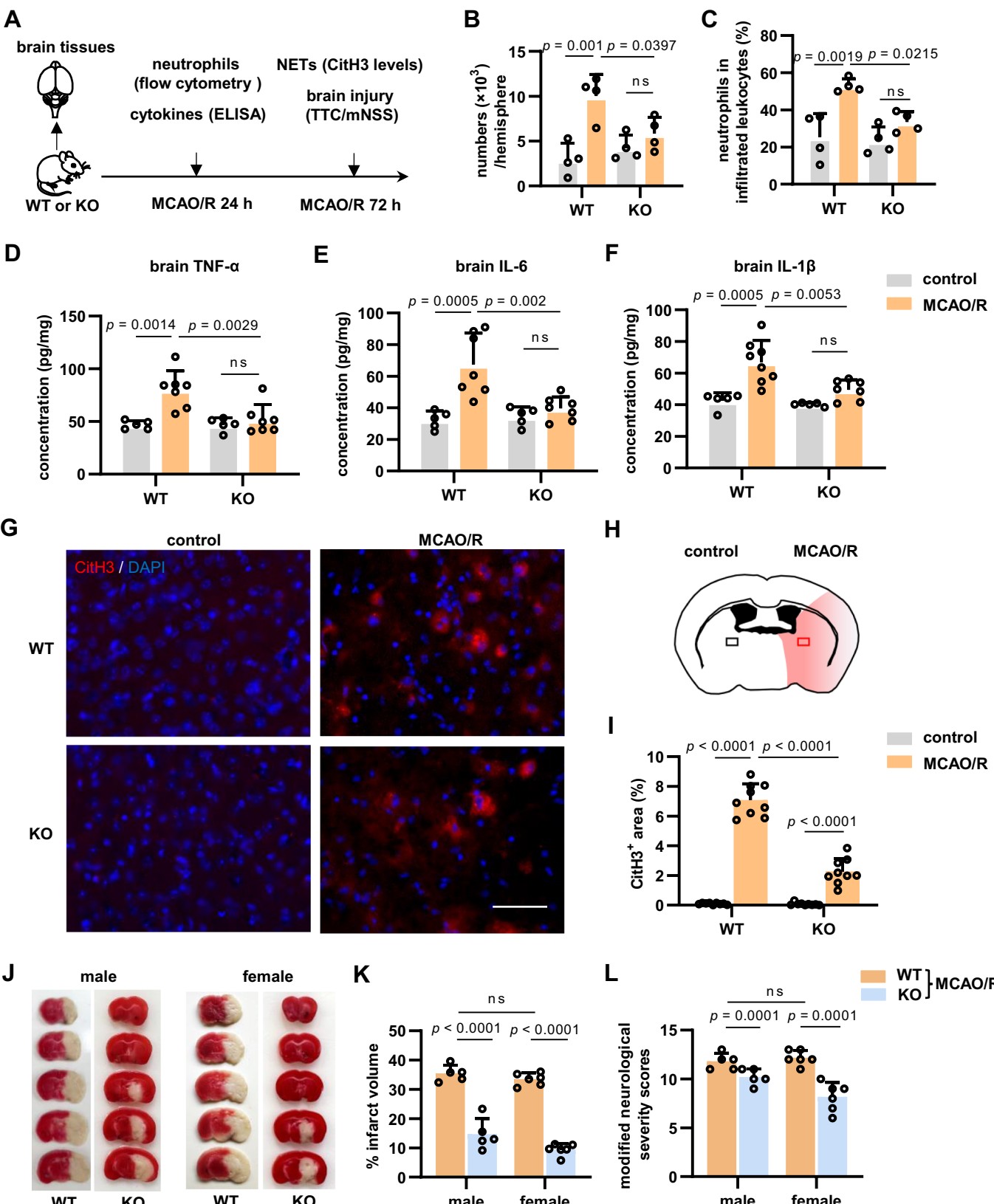

**Figure 6.   HPK1 loss ameliorates brain inflammation and injury in mice after ischemic stroke.**

(A) Schematic showing the experimental procedures for assessing brain inflammation and injury of WT/KO mice after MCAO/R. (B, C) Flow cytometry analysis of the number (B) and frequency (C) of infiltrated neutrophils in the brain tissues after sham or MCAO/R for 24 h. $N = 4$ mice per group. (D–F) ELISA assay of TNF-α (D), IL-6 (E), and IL-1β (F) levels in the brain tissues after sham or MCAO/R for 24 h. $N = 5$–8 mice per group. (G–I) Representative CitH3 immunofluorescence images (G), schematic diagram of CitH3$^+$ area (H), and quantification (I) in the brain slices after MCAO/R for 72 h. Scale bar = 50 μm. $N = 9$ per group, including nine sections from three mice. (J, K) Representative TTC staining (J) and infarct volume (K) of male and female mice after MCAO/R for 72 h, $N = 5$–6 mice per group. (L) The mNSS evaluation of male and female mice after MCAO/R for 72 h. $N = 5$–6 mice per group. Bar graphs represent the mean ± SD. ns, no significance. Two-way ANOVA in (B–F), (I), (K), and (L). Source data are available online for this figure.

ischemic hemisphere after MCAO (90 min)/R for 24 h (Fig. 8G–I). These results demonstrate that i-HPK1 treatment ameliorates brain injury and neuroinflammation after ischemic stroke, highlighting the potential of i-HPK1 in AIS treatment.

# Discussion

Systemic inflammation predisposes patients to ischemic stroke and induces pathogenic factors detrimental to brain recovery after reperfusion. Circulating neutrophil responses are associated with stroke severity and functional outcomes (Jickling et al, 2015; Laridan et al, 2017; Denorme et al, 2022). Given their important roles in ischemic stroke, neutrophils have emerged as potential therapeutic targets. However, targeting neutrophil adhesion has failed in clinical trials because of unfavorable outcomes (Elkind et al, 2020). We speculated that targeting a single neutrophil function would not be efficient in eliciting a satisfactory outcome. Searching for molecular targets that can simultaneously mediate multiple neutrophil responses is a potential strategy. In this study, we revealed that HPK1 mediates neutrophil mobilization, hyperactivation involving inflammatory responses, NETosis, and tissue infiltration following AIS. Importantly, both HPK1 genetic deletion and inhibition ameliorate pulmonary and cerebral reperfusion injuries in mice within the acute phase following AIS. Our findings highlight HPK1 as a novel therapeutic target for improving post-stroke outcomes.

Compared with its roles in adaptive immunity and tumor immunosuppression (Chuang et al, 2016; Hernandez et al, 2018; Si et al, 2020; Sun et al, 2023), the functional impact of HPK1 on innate immunity has been ignored. A previous study has revealed that HPK1 positively regulates neutrophil adhesion following acute inflammation (Jakob et al, 2013). In this study, we found that neutrophil-intrinsic HPK1 enhances proinflammatory responses, ROS production, and NET formation, involving NF-κB/STAT3/p38-MAPK and GSDMD signaling pathways. NF-κB is a master mediator of inflammatory responses and inflammasome regulation in myeloid cells (Futosi et al, 2013; Liu et al, 2017). GSDMD is observed within canonical NLRP3 inflammasomes (He et al, 2015). In neutrophils, canonical inflammasomes mediate the cleavage of GSDMD, which is required for proinflammatory factor release in a lytic-independent manner (Heilig et al, 2018; Karmakar et al, 2020). During NETosis induction, cleaved GSDMD forms pores in the plasma membrane, thereby facilitating NET release (Sollberger et al, 2018). STAT3 signal pathway controls bone marrow neutrophil mobilization by upregulating CXCR2 expression and neutrophil trafficking during acute inflammation (Fielding et al, 2008; Nguyen-Jackson et al, 2010). Furthermore, the p38-MAPK pathway is involved in neutrophil trafficking, inflammatory and

oxidative responses, and neutrophil degranulation (Kim and Haynes, 2013; Senger et al, 2017; Zhou et al, 2022). ROS mediates p38-MAPK activation and then NET release from human neutrophils (Keshari et al, 2013). In addition, p38-MAPK phosphorylates and activates canonical inflammasome (Rodríguez-Ruiz et al, 2023). Above studies suggest that HPK1 aggravates neutrophil responses and NETosis probably by regulating the crosstalk between NF-κB/STAT3/p38-MAPK and GSDMD signaling pathways.

Furthermore, this study revealed that HPK1 mediates neutrophil mobilization and systemic inflammation in mice after AIS and reperfusion. Patients with AIS exhibit high NLR during the acute phase, which predicts post-stroke respiratory morbidity, including early pneumonia, hemorrhagic development, and poor functional outcomes (Zhang et al, 2017; Nam et al, 2018a; Nam et al, 2018b; Zhang et al, 2019; Bi et al, 2021). NET formation consistently indicates stroke severity and poorer prognosis in patients with AIS (Wu et al, 2024). However, the pathological significance and underlying mechanism of rapid neutrophil responses following stroke remain poorly understood. Our data prove that HPK1 increases neutrophil CXCR2 levels, contributing to neutrophil mobilization after stroke. The STAT3 signaling pathway regulates CXCR2 expression in the bone marrow during mobilization (Nguyen-Jackson et al, 2010). Therefore, HPK1 is involved in CXCR2 expression, accounting for neutrophil mobilization into the blood and infiltration into the brain and lungs after stroke.

The lungs are particularly vulnerable to AIS. Many patients develop acute lung injury during the early phases of AIS (Bai et al, 2018; Wang et al, 2022). Immunosuppression within the sub-acute phase has been implicated in pulmonary infection and dysfunction. This study showed that AIS leads to acute inflammation state and lung injury in mice, consistent with previous reports (Samary et al, 2018; Xu et al, 2022). Neutrophil infiltration and NETs are upregulated in post-stroke lung tissues. NET formation is highly prevalent in LPS- and mechanical ventilation-induced lung inflammation and injury (Li et al, 2017; Shi et al, 2020; Yildiz et al, 2015). Furthermore, we found that targeting HPK1 ameliorates neutrophil infiltration and NET formation in lung tissues, alleviating lung injury and inflammation after ischemic stroke. Our study suggests that targeting HPK1 is a potential strategy for treating post-stroke lung complications.

Small-molecule inhibitors targeting HPK1 provide antitumor immunity responsiveness (Si et al, 2020; Linney and Kaila, 2021; Zhou et al, 2022); however, their therapeutic effects on acute brain disorders have not been revealed. Our study provides new evidence that i-HPK1 suppresses neutrophil hyperactivation by blocking NF-κB/STAT3/p38-MAPK and GSDMD pathways and neutrophil trafficking into the brain and lungs, ultimately alleviating cerebral

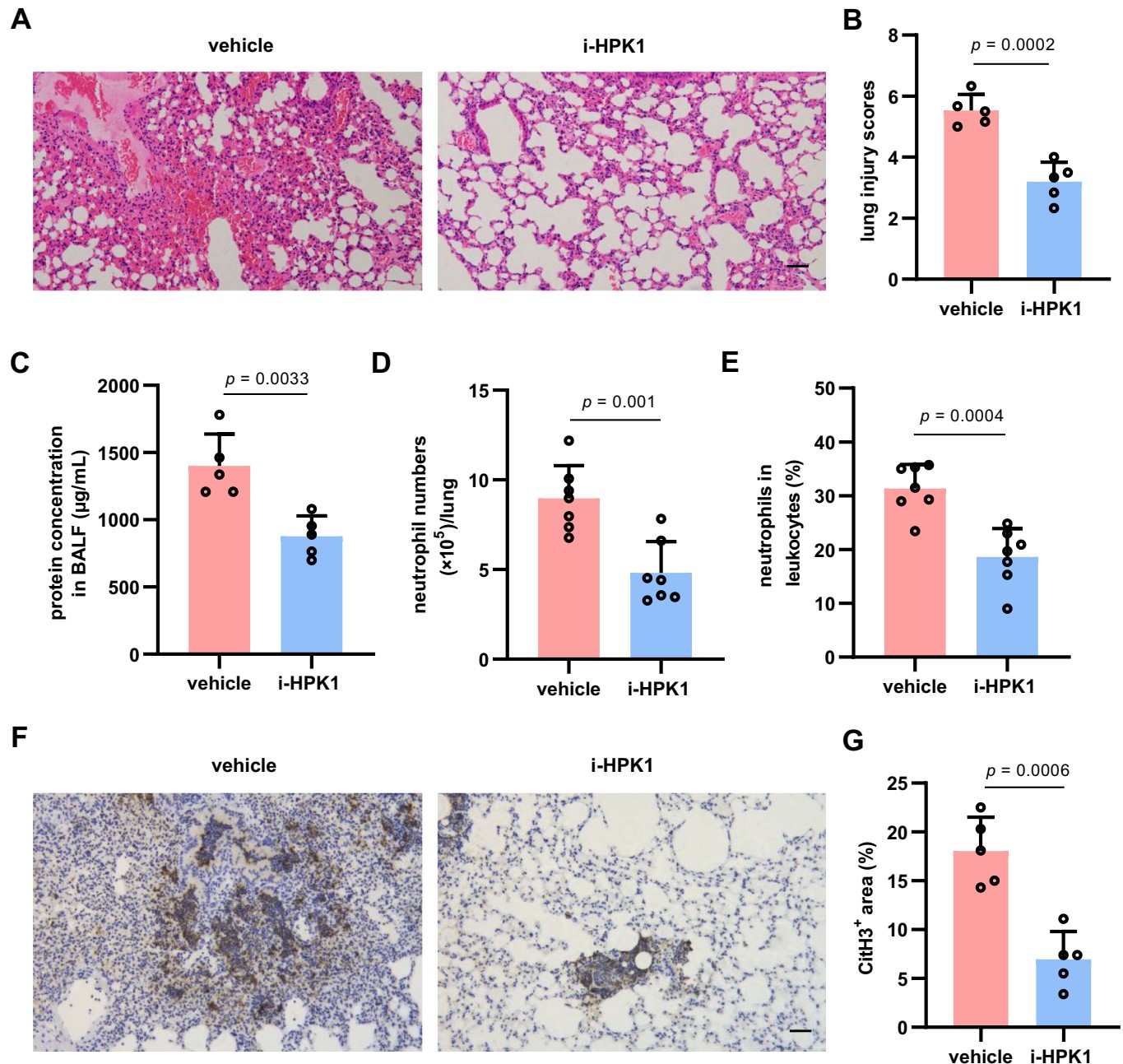

**Figure 7. Inhibiting HPK1 ameliorates lung injury and inflammation in mice after ischemic stroke.**

(A, B) HE staining (A) and histopathological scores (B) of mouse lung sections after MCAO/R for 48 h. Scale bar = 50 μm. N = 5 mice per group. (C) Protein concentration of BALF after MCAO/R for 48 h. N = 5 mice per group. (D, E) Flow cytometry analysis of the number (D) and frequency (E) of infiltrated neutrophils in mouse lung tissues after MCAO/R for 24 h. N = 7 mice per group. (F, G) Representative CitH3 immunohistochemical images (F) and quantification (G) of in mouse lung sections after MCAO/R for 48 h. N = 5 mice per group. Mice were treated with vehicle or i-HPK1 30 min after reperfusion following MCAO (90 min). Scale bar = 50 μm. Bar graphs represent the mean ± SD. Unpaired Student's t-test in (B–E) and (G). Source data are available online for this figure.

infarction and pulmonary injury following AIS. i-HPK1 has demonstrated high selectivity and efficiency in pre-clinical studies and has proven safe in clinical studies (Deva et al, 2024), highlighting its clinical translation potential in AIS treatment.

A key strength of this study is the determination of the roles of HPK1 in neutrophil mobilization and systemic inflammation

following AIS, validating a potential strategy for AIS treatment. A significant limitation of this study is the absence of data from patients with AIS. Further studies obtaining patient data from peripheral blood will benefit i-HPK1 clinical translation. In addition, HPK1 is expressed in macrophages and lymphocytes. Further studies are required to investigate HPK1 roles in specific

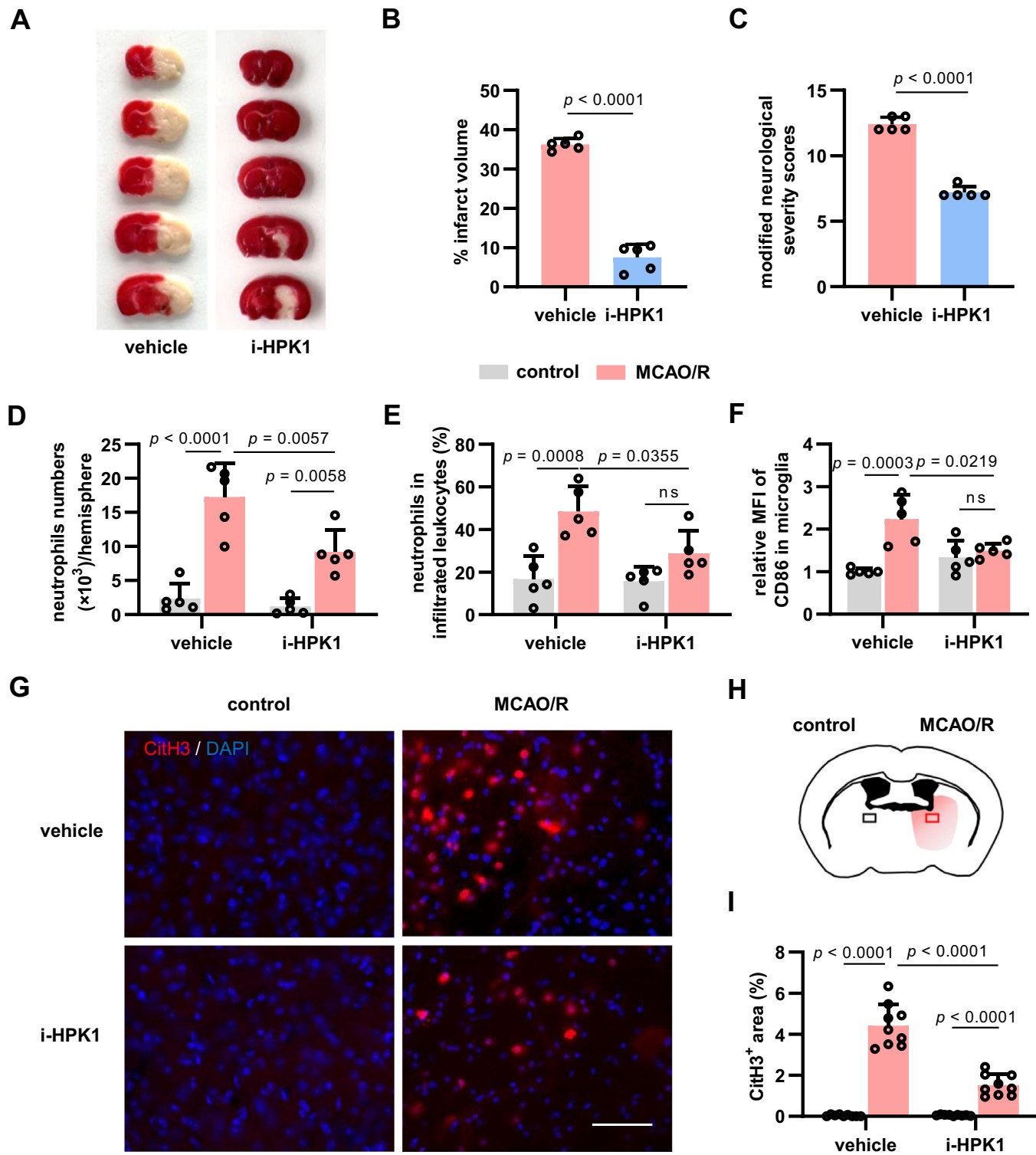

**Figure 8. Inhibiting HPK1 ameliorates brain injury and inflammation in mice after ischemic stroke.**

(A, B) Representative TTC staining (A) and infarct volume (B) after MCAO/R for 48 h. $N = 5$ mice per group. (C) The mNSS evaluation after MCAO/R for 48 h. $N = 5$ mice per group. (D, E) Flow cytometry analysis of the number (D) and frequency (E) of infiltrated neutrophils in mouse brain tissues after MCAO/R for 24 h. $N = 5$ mice per group. (F) Flow cytometry analysis of CD86 expression (MFI) in microglia after MCAO/R for 24 h. $N = 5$ mice per group. (G–I) Representative CitH3 IF images (G), schematic diagram of CitH3+ area (H), and quantification (I) in mouse brain slices after MCAO/R for 24 h. Scale bar = 50 μm. $N = 9$ per group, including nine sections from three mice. Mice were treated with vehicle or i-HPK1 30 min after reperfusion following MCAO (90 min). Bar graphs represent the mean ± SD. ns, no significance. Unpaired Student's $t$-test in (B) and (C), two-way ANOVA in (D–F) and (I). Source data are available online for this figure.

leukocytes following AIS. Behavioral experiments are also essential to evaluate the role of HPK1 in long-term outcomes. In summary, we demonstrated the role of HPK1 in regulating neutrophil activation and mobilization, providing a therapeutic target for post-stroke brain and lung injuries.

# Methods

### Regents and tools table

| Reagent/Resource | Reference or Source | Identifier or Catalog Number |
|---|---|---|
| **Experimental models** | | |
| *Map4k1*$^{+/-}$ heterozygous mice | Texas A&M Institute for Genomic Medicine | |
| C57BL/6 | Cavens Lab Animal Ltd | |
| **Antibodies** | | |
| Rabbit anti-HPK1 | Cell Signaling Technology | 4472 |
| Rabbit anti-phospho-NF-κB p65 | Abcam | ab76302 |
| Rabbit anti-NF-κB p65 | Proteintech | 10745-1-AP |
| Rabbit anti-phospho-IκBα | Proteintech | 82349-1-RR |
| Rabbit anti-IκBα | Abmart | T55026 |
| Rabbit anti-phospho-STAT3 | Cell Signaling Technology | 9145 |
| Rabbit anti-STAT3 | Proteintech | 10253-2-AP |
| Rabbit anti-phospho-p38 | Proteintech | 28796-1-AP |
| Rabbit anti-p38 | Proteintech | 14064-1-AP |
| Rabbit anti-GSDMD | Zen-bioscience | R40136 |
| Rabbit anti-CXCR2 | Proteintech | 20634-1-AP |
| Mouse anti-β-actin | Cell Signaling Technology | 3700 |
| Horse anti-mouse IgG | Cell Signaling Technology | 7076 |
| Goat anti-rabbit IgG | Cell Signaling Technology | 7074 |
| Rabbit anti-CitH3 | Abcam | ab281584 |
| Horseradish peroxidase-conjugated goat anti-rabbit IgG | Servicebio | GB23303 |
| Alexa Fluor 594 goat anti-rabbit IgG | Invitrogen | A-11012 |
| Rat anti-mouse CD16/CD32 | BD Bioscience | 553141 |
| APC anti-mouse CD45 | Biolegend | 103112 |
| Percp cy5.5 anti-mouse CD11b | Invitrogen | 45-0112-82 |
| PE anti-mouse Ly6G | Invitrogen | 12-9668-82 |
| FITC anti-mouse CXCR2 | Biolegend | 149309 |
| APC anti-mouse CXCR4 | Invitrogen | 17-9991-82 |
| FITC anti-mouse CD86 | Biolegend | 105005 |
| **Oligonucleotides and other sequence-based reagents** | | |
| PCR primers | Sangon Biotech | Appendix Table S1 |

| Reagent/Resource | Reference or Source | Identifier or Catalog Number |
|---|---|---|
| **Chemicals, Enzymes and other reagents** | | |
| Mouse Peripheral Blood Neutrophil Separation Kit | TBD Science | LZS1100 |
| Histopaque 1119 | Sigma-Aldrich | 11191 |
| Histopaque 1077 | Sigma-Aldrich | 10771 |
| Percoll | Solarbio | P8370 |
| RPMI 1640 medium | Gibco | C11875500BT |
| Fetal bovine serum | ExCell Bio | FSP500 |
| Penicillin-Streptomycin | Gibco | 15140122 |
| LPS | Sigma-Aldrich | L2630 |
| Nigericin | Yuanye Bio | S25116 |
| i-HPK1 | Selleck | E1297 |
| SYTOX Green | Invitrogen | S7020 |
| Liberase TL | Roche | 5401020001 |
| DNase I | Beyotime | D7073 |
| Trizol reagent | Vazyme | R411 |
| HiScript III RT SuperMix | Vazyme | R323 |
| ChamQ SYBR qPCR Master Mix | Vazyme | Q311 |
| RIPA lysis buffer | Beyotime | P0013 |
| protease inhibitor cocktail | APexBio | K1011 |
| Reactive oxygen species assay kit | NJJC Bio | E004 |
| Mouse TNF-α ELISA kit | ABclonal | RK00027 |
| Mouse IL-6 ELISA kit | ABclonal | RK00008 |
| Mouse IL-1β ELISA kit | ABclonal | RK00006 |
| Enhanced ECL Chemiluminescence Detection Kit | Vazyme | E422-02 |
| 2,3,5-triphenyitetrazolium chloride | Solarbio | T8170 |
| **Software** | | |
| ImageJ | https://imagej.nih.gov/ij/index.html | |
| GraphPad Prism 8.0 | https://www.graphpad.com | |
| Flowjo 10.0.7r2 | https://www.flowjo.com | |

## Methods and protocols

### Animals

Adult C57BL/6 and *Map4k1*$^{+/-}$ heterozygous mice were purchased from Cavens Lab Animal Ltd. (Changzhou, Jiangsu, China) and Texas A&M Institute for Genomic Medicine (College Station, TX, USA), respectively. The mice were housed under a 12-h light-dark cycle with free access to food and water. Adult male mice were used except where both sexes were indicated. All animal experiments

were approved by the local Animal Care and Use Committee (approval number: IACUC-2307025).

## Primary bone marrow neutrophil isolation and culture

Mouse bone marrow neutrophils were isolated using the Histopaque 1077/1119 methods as previously described (Swamydas and Lionakis, 2013). Briefly, bone marrow cells from the femurs and tibias were flushed with RPMI 1640 medium (Gibco), supplemented with 10% fetal bovine serum (ExCell Bio) and 1% penicillin-streptomycin (Gibco), and then bone marrow cells were filtered through a sterile 70-μm nylon mesh cell strainer. After lysis of the red blood cells with 0.2% NaCl, bone marrow cells were laid on the Histopaque 1077 and 1119 (Sigma-Aldrich) and subsequently centrifuged at $500 \times g$ for 30 min. Neutrophils were collected between Histopaque 1077 and 1119. Neutrophil purity was analyzed using flow cytometry, which was always >90%. After 30-min incubation in the complete culture medium at 37 °C, neutrophils were stimulated with LPS (Sigma-Aldrich). Nigericin (10 μM, Yuanye Bio) was treated for 45 min following LPS (500 ng/mL) priming for 3 h. i-HPK1 ((R)-4-(5-amino-6-((1-(1-methylpiperidin-4-yl)-1H-pyrazol-4-yl)oxy)pyrazin-2-yl)-N-(2-(3-fluoropyrrolidin-1-yl)ethyl)-2,6-dimethylbenzamide) (2 μM, Selleck) was added 30 min before LPS stimulation. Data were obtained from at least three independent cultures. The primary neutrophils were free of mycoplasma contamination (GMyc-PCR Mycoplasma Test Kit, Yeasen).

## Quantitative PCR (qPCR)

The mRNA levels of *Tnf*, *Il6*, *Il1b*, *Ptgs2, Socs3*, and *Il10* in primary neutrophils were analyzed using qPCR after LPS stimulation for 1 h. *Nos2* mRNA level was analyzed after LPS stimulation for 3 h. Total RNA in primary neutrophils was extracted using Trizol and cDNA was synthesized using HiScript III RT SuperMix per manufacturer instructions (Vazyme). qPCR was performed on an ABI QuantStudio3 System (Thermo Fisher Scientific, Waltham, MA, USA) using the ChamQ SYBR qPCR Master Mix (Vazyme). Primers are listed in Appendix Table S1. Double delta CT log2 was calculated and the data are presented as fold change.

## SYTOX Green assay

Membrane-impermeable SYTOX Green fluorescent dye (Invitrogen) was used to mark neutrophils undergoing NETosis and extracellular NETs. Mouse peripheral blood neutrophils were isolated using an isolation kit (TBD Science) per manufacturer instructions. Regarding the NET formation assay, neutrophils from the bone marrow or peripheral blood were seeded on poly-lysine-coated coverslips ($5 \times 10^5$ cells/coverslip), stimulated with LPS for 2.5 h, and then incubated with SYTOX Green (1:1000, Invitrogen) for 15 min. The staining was visualized under a MF52-N inverted fluorescence system (Mshot, Guangzhou, Guangdong, China). For quantification, the SYTOX Green$^+$ area fraction was measured in three random fields using ImageJ software.

## MCAO model and drug treatment

Adult mice were subjected to MCAO (60 or 90min) as previously described (Xia et al, 2024). Briefly, the WT and HPK1-KO mice were anesthetized with 3% isoflurane for induction and 1% for maintenance throughout surgery. A silicone-coated filament (Getimes, Beijing, China) was directed to the middle cerebral artery (MCA) for 60 or 90-min occlusion of MCA. i-HPK1 (2 mg/

kg) was administrated once daily via tail intravenous injection 30 min after MCAO/R.

## Neurological scores and infarct volume analysis

Neurological dysfunction was assessed after MCAO/R for 48 h or 72 h using an mNSS system, including motor, sensory, balance, and reflex tests. Neurological performance was evaluated in a blinded manner by an experienced investigator.

For infarct volume analysis, the brains were removed and frozen at −20 °C for 20 min, then sliced into five slices with 1-mm thickness and stained with 2,3,5-triphenyitetrazolium chloride (Solarbio) for 10 min at 37 °C, and fixed in 4% paraformaldehyde overnight. Brain infarct volumes were measured using the ImageJ software. Infarct volume (%) = (the volume of control hemisphere − the non-infarcted volume of MCAO/R hemisphere)/(2 × the volume of control hemisphere) × 100%.

## Hematoxylin and eosin (HE) staining

Lung tissues were collected after MCAO/R for 48 h or 72 h, cut into 5-μm sections, and stained with haematoxylin and eosin. Lung injury severity was evaluated in three random fields, as described previously (Song et al, 2019; Kulkarni et al, 2022). The histological scores included four items: alveolar edema, hemorrhage, alveolar septal thickening, and leukocyte infiltration. Each item was assessed on a scale ranging from 0 to 3 (0 = normal; 1 = mild; 2 = moderate; 3 = severe), and the total acute lung injury score was then calculated.

## Determination of protein concentration in BALF

Lung tissues were laminated with phosphate-buffered saline (PBS) to collect BALF, which was centrifuged at $1500 \times g$ for 10 min at 4 °C. The protein content in the supernatant was determined using the Lowry method.

## Flow cytometry

Blood samples were collected in ethylenediaminetetraacetic acid (EDTA)-coated collection tubes. Bone marrow samples were flushed with 1640 complete culture medium and subsequently filtered through a sterile 70-μm cell strainer. Spleen samples were homogenized and pressed through a sterile 70-μm cell strainer. Red blood cells were lysed in 0.2% NaCl. Brain hemispheres were removed, freed from meninges, and gently shredded in a small amount of ice-cold PBS. Brain tissues were incubated for 60 min at 37 °C in PBS containing 0.25 mg/mL Liberase TL (Roche) and 50 U/mL DNase I (Beyotime), then passed through a sterile 70-μm cell strainer. Myelin debris was separated using a 40% Percoll solution (Solarbio). The pelleted cells were collected for further staining. Lung tissues were cut into small pieces, incubated for 60 min at 37 °C in PBS containing 0.25 mg/mL Liberase TL and 10 μg/mL DNase I, and passed through a sterile 70-μm cell strainer.

Single cells from blood or various tissues were incubated at 4 °C for 15 min with CD16/CD32 (1:200, BD Bioscience) and then incubated with antibodies for 30 min at 4 °C. The following antibodies were used: CD45-APC (1:200, Biolegend), CD11b-Percp cy5.5 (1:100, Invitrogen), Ly6G-PE (1:100, Invitrogen), CXCR2-FITC (1:100, Biolegend), CXCR4-APC (1:100, Invitrogen), and CD86-FITC (1:200, Biolegend). For ROS detection, bone marrow and peripheral blood neutrophils were incubated with DCFH-DA (1:2000, NJJC Bio) for 15 min at 37 °C. Stained cells were washed and resuspended in 300 μL of PBS and the whole suspension was acquired using a FACSCalibur flow

cytometer (BD, Franklin Lakes, NJ, USA). Data were analyzed using Flowjo 10.0.7r2 software. The gating strategies for flow cytometry analysis are shown in Appendix Fig. S1. Simply described, neutrophil populations in the blood, bone marrow, lung, and spleen were set as follows: First, debris was excluded from single cells. Next, leukocytes were gated as CD45$^+$ cells. Finally, neutrophils were gated as CD11b$^+$ and Ly6G$^+$ cells from the CD45$^+$ cells (Appendix Fig. S1A). Brain-infiltrated neutrophils were gated as CD11b$^+$ and Ly6G$^+$ cells from the CD45$^{high}$ population, while microglia were gated as CD45$^{intermediate}$ and CD11b$^+$ cells (Appendix Fig. S1B).

### Enzyme-linked immunosorbent assay (ELISA)

Isolated EDTA-treated blood was centrifuged at $1000 \times g$ at room temperature for 10 min to obtain the plasma. Brain and lung tissue samples were homogenized with PBS containing protease inhibitor (ApexBio) and centrifuged at $5000 \times g$ at 4 °C for 15 min. The supernatants and plasma were stored at $-80$ °C. Plasma and supernatants from brain and lung homogenates were used to determine cytokine concentration with mouse ELISA kits (ABclonal) per manufacturer instructions.

### Immunoblotting

Bone marrow and peripheral blood neutrophils were lysed in RIPA buffer (Beyotime) containing protease inhibitor (ApexBio). The gel electrophoresis and transfer process were performed as previously described (Sun et al, 2023). The primary antibodies used were: HPK1 (1:1000, Cell Signaling Technology), phospho-NF-κB p65 (1:4000, Abcam), NF-κB p65 (1:4000, Proteintech), phospho-STAT3 (1:1000, Cell Signaling Technology), STAT3 (1:1000, Proteintech), phospho-IκBα (1:1000, Proteintech), IκBα (1:1000, Abmart), phospho-p38-MAPK (1:4000, Proteintech), p38-MAPK (1:4000, Proteintech), CXCR2 (1:2000, Proteintech), GSDMD (1:1000, Zen-bioscience), and β-actin (1:10,000, Cell Signaling Technology). Specific proteins were detected using an enhanced ECL Chemiluminescence Detection Kit (Vazyme) and an Amersham ImageQuant 800 imaging system (Cytiva, Marlborough, MA, USA). Quantitative analysis was performed using ImageJ software.

### Immunohistochemistry

Lung tissues were collected and cut into 5-μm sections. The sections were incubated with rabbit anti-CitH3 antibody (1:200, Abcam) overnight at 4 °C, and then washed in PBS. The sections were incubated with a horseradish peroxidase-conjugated goat anti-rabbit IgG (1:500, Servicebio) for 30 min at room temperature. Finally, the sections were stained with 3-3'-diaminobenzidine and counterstained with hematoxylin. For quantification, the CitH3$^+$ area fraction was measured in three random fields using ImageJ software.

### CitH3 IF

The brains were removed, and 30-μm sections were prepared using a CM3050S freezing microtome (Leica, Wetlzlar, Hesse, Germany). Sections were blocked with 1% bovine serum albumin, 0.25% Triton X-100, and 10% goat serum in PBS and were incubated with rabbit anti-CitH3 (1:500, Abcam) overnight at 4 °C. After washing with PBS, sections were incubated with Alexa Fluor 594-conjugated goat anti-rabbit IgG (1:500, Invitrogen) in PBS for 1 h at room temperature. For quantification, the CitH3$^+$ area fraction was measured in three random fields using ImageJ software.

### The paper explained

**Problem**

An increase in the number of circulating neutrophils correlates with poor neurological outcomes and respiratory morbidity after ischemic stroke. However, the molecular mechanisms underlying neutrophil mobilization and the various responses remain unclear; moreover, the pathological impact and regulation of neutrophils in post-stroke lung injury have been overlooked.

**Results**

HPK1 protein was upregulated after LPS stimulation in bone marrow-derived neutrophils, and HPK1 deletion mitigated LPS-induced neutrophil hyperactivation. Moreover, HPK1 loss inhibited neutrophil mobilization and hyperactivation of peripheral and infiltrated neutrophils and alleviated post-stroke systemic inflammation in mice. Both genetic knockout and pharmacological inhibition of HPK1 progressively alleviated lung and cerebral injuries after ischemic stroke.

**Impact**

Our study determined the roles of HPK1 in neutrophil mobilization and responses after ischemic stroke and identified new treatments for brain and lung injuries following ischemic stroke. This study highlights HPK1 as a novel therapeutic target for improving post-stroke outcomes.

### Statistical analyses

Mice were randomly grouped into sham and MCAO/R groups. The sample size is indicated in the Figure legends. All data are presented as the mean ± standard deviation (SD). The statistical analyses were performed using the GraphPad Prism 8.0 software. Unpaired two-tailed Student's $t$-test was used for analysis between two groups with one variable. Comparisons among multiple groups were assessed using one-way analysis of variance (ANOVA) followed by Dunnett's comparison test, two-way ANOVA followed by Tukey's or Bonferroni's multiple comparison test. A value of $P < 0.05$ was considered statistically significant. Outliers were identified using the Grubbs outlier test and excluded from the analysis.

## Data availability

This study includes no data deposited in external repositories.

The source data of this paper are collected in the following database record: biostudies:S-SCDT-10_1038-S44321-025-00220-8.

## Peer review information

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

## Acknowledgements

The authors greatly acknowledge the financial support of the grants from the National Natural Science Foundation of China (82173801 and 81673418) and the Natural Science Foundation of the Jiangsu Higher Education Institutions of China (18KJA310007).

## Author contributions

**Tingting Zhang**: Formal analysis; Investigation; Visualization; Methodology; Writing—original draft. **Ying Sun**: Formal analysis; Investigation; Visualization; Methodology; Writing—original draft. **Jing Xia**: Methodology. **Hongye Fan**: Resources; Validation; Methodology. **Dingfang Shi**: Resources; Validation; Methodology. **Qian Wu**: Resources; Supervision. **Ming Huang**: Resources; Supervision; Methodology. **Xiao-Yu Hou**: Conceptualization; Supervision; Funding acquisition; Methodology; Project administration; Writing—review and editing.

Source data underlying figure panels in this paper may have individual authorship assigned. Where available, figure panel/source data authorship is listed in the following database record: biostudies:S-SCDT-10_1038-S44321-025-00220-8.

## Disclosure and competing interests statement

The authors declare no competing interests.

# Expanded View Figures

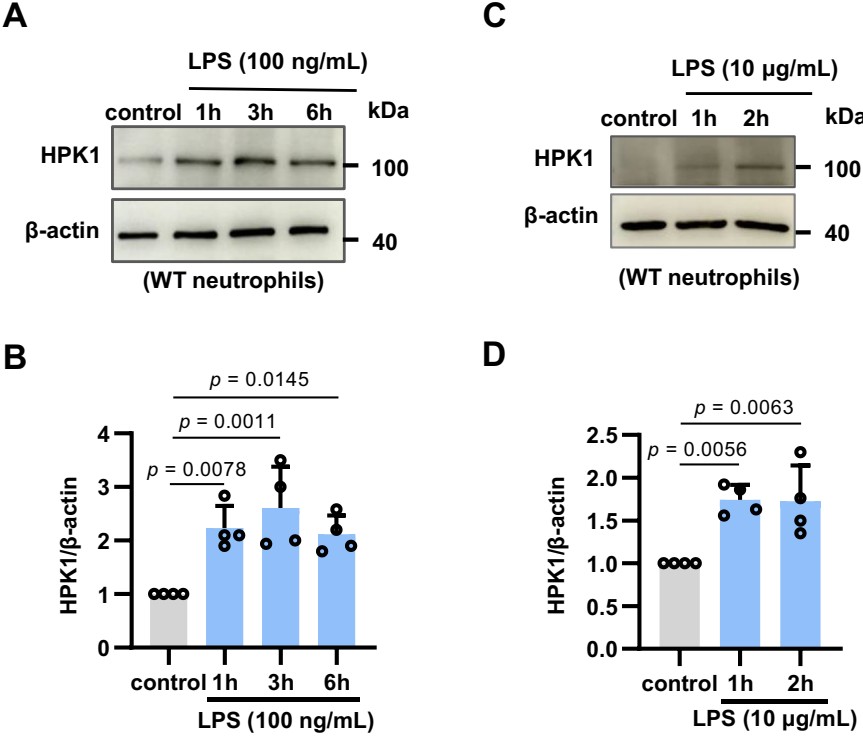

**Figure EV1.  HPK1 levels increase in primary neutrophils after LPS treatment.**

(**A, B**) Representative HPK1 immunoblot (**A**) and quantification (**B**) in primary neutrophils from WT mice treated with LPS (100 ng/mL) for 1, 3, and 6 h. N = 4 per group. (**C, D**) Representative HPK1 immunoblot (**C**) and quantification (**D**) in primary neutrophils from WT mice treated with LPS (10 µg/mL) for 1 and 2 h. N = 4 per group. Bar graphs represent the mean ± SD. One-way ANOVA in (**B**) and (**D**). Source data are available online for this figure.

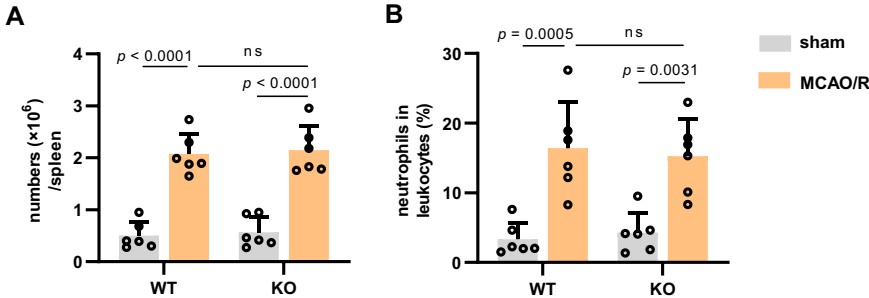

**Figure EV2.  HPK1 loss has no influence on neutrophil population in the mouse spleen after ischemic stroke.**

(**A**, **B**) Flow cytometry analysis of the number (**A**) and frequency (**B**) of neutrophils in the spleen tissues of WT/KO mice after sham or MCAO/R for 24 h. $N = 6$ mice per group. Bar graphs represent the mean ± SD. ns, no significance. Two-way ANOVA in (**A**) and (**B**). Source data are available online for this figure.

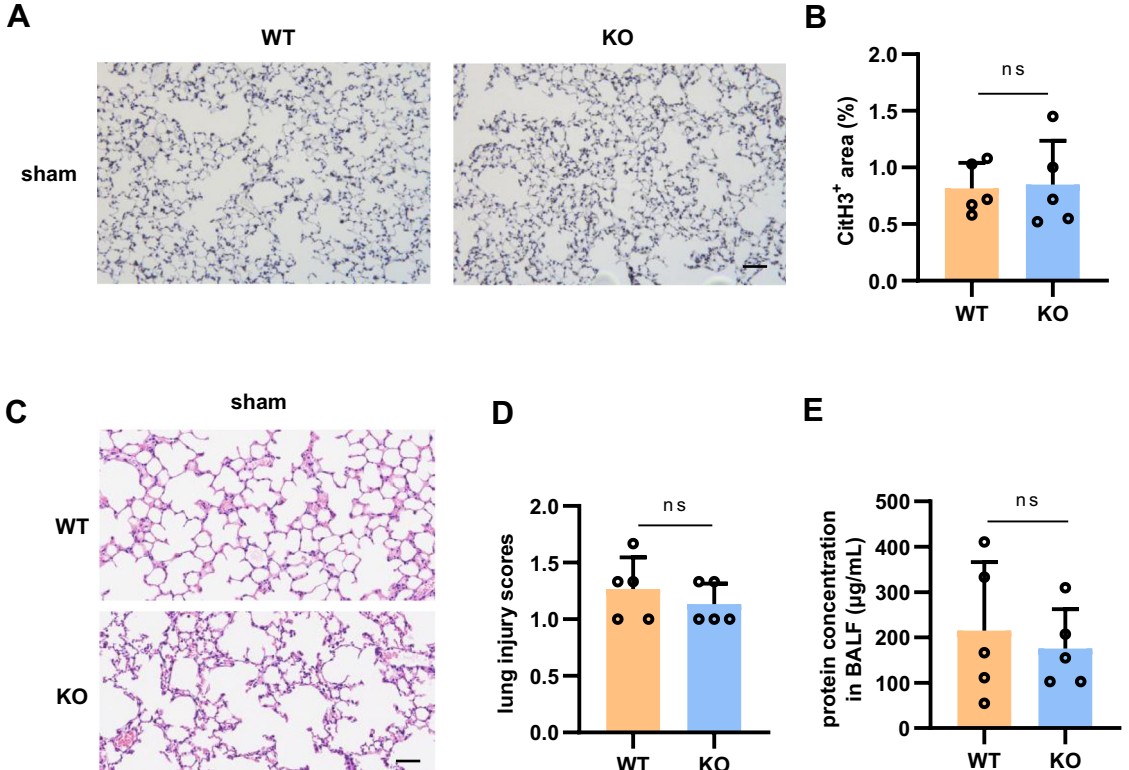

**Figure EV3.  There are no marked NETs and lung injury in WT and HPK1 loss mice after sham operation.**

(**A, B**) Representative CitH3 immunohistochemical images (**A**) and quantification (**B**) in the lung sections from WT/KO mice after sham operation. $N = 5$ mice per group. Scale bar = 50 μm. (**C, D**) Representative HE staining (**C**) and histopathological scores (**D**) of lung sections from WT/KO mice after sham operation. Scale bar = 50 μm. $N = 5$ mice per group. (**E**) Protein concentration of BALF in WT/KO mice after sham operation. $N = 5$ mice per group. Bar graphs represent the mean ± SD. ns, no significance. Unpaired Student's *t*-test in (**B**), (**D**), and (**E**). Source data are available online for this figure.

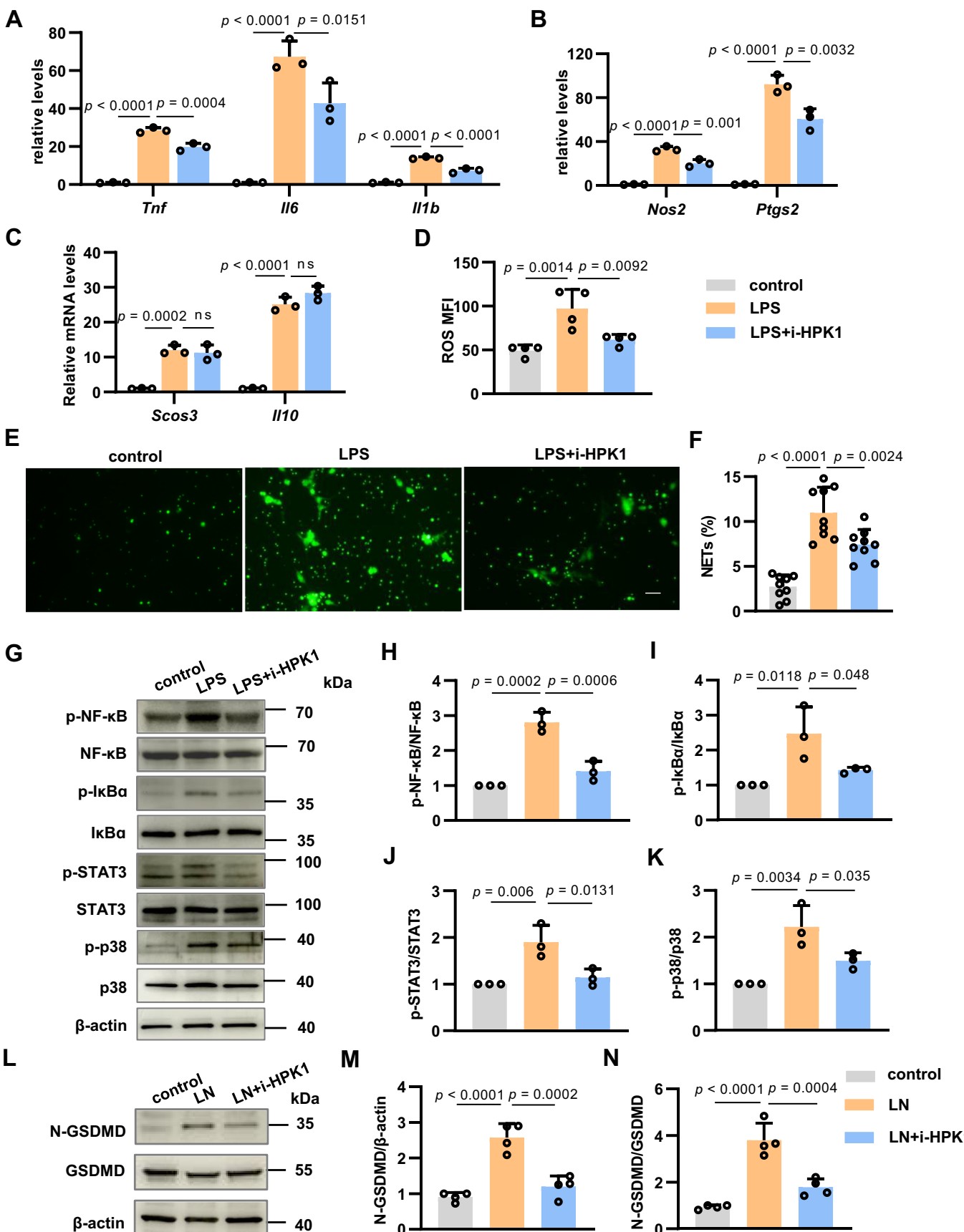

◄ **Figure EV4. Inhibiting HPK1 reduces LPS-induced hyperactivity and NF-κB/STAT3/p38/GSDMD pathways in primary neutrophils.**

(A–C) qPCR analysis of *Tnf*, *Il6*, and *Il1b* (A), *Nos2* and *Ptgs2* (B), *Socs3* and *Il10* (C) in WT neutrophils treated with LPS (100 ng/mL). $N = 3$ per group, three-time repeats. (D) Flow cytometry analysis of ROS levels (MFI) in WT neutrophils treated with LPS (100 ng/mL) for 3 h. $N = 4$ per group, three-time repeats. (E–F) Representative fluorescence imaging (E) and quantification (F) of NET formation. Neutrophils were stimulated with LPS (10 μg/mL) for 2.5 h, and NETs were visualized using SYTOX Green. Scale bar $= 50$ μm. $N = 9$ per group. (G–K) Representative immunoblot (G) of phosphorylation and total protein levels of NF-κB (H), IκBα (I), STAT3 (J), and p38 (K) in WT neutrophils treated with LPS (100 ng/mL). $N = 3$ per group. (L–N) Representative immunoblot (L) and quantification of GSDMD cleavage (N-GSDMD) levels (M) and N-GSDMD/GSDMD ratio (N) in WT neutrophils treated with LPS (500 ng/mL) and nigericin (LN). $N = 4$ per group. i-HPK1 was treated 30 min before LPS treatment. Bar graphs represent the mean ± SD. ns, no significance. One-way ANOVA in (A–D), (F), (H–K), (M), and (N). Source data are available online for this figure.

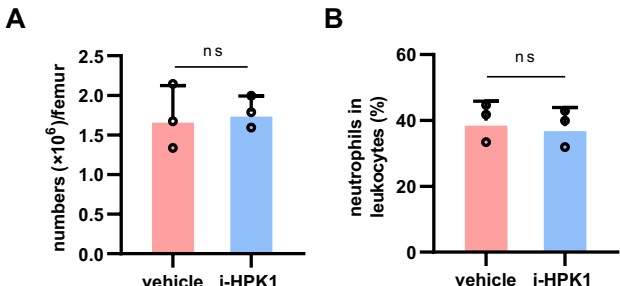

**Figure EV5. Inhibition of HPK1 has no influence on neutrophil release from bone marrow in mice after ischemic stroke.**

(A, B) Flow cytometry analysis of the number (A) and frequency (B) of bone marrow neutrophils in mice after MCAO/R for 24 h. Mice were treated with vehicle or i-HPK1 30 min after reperfusion following MCAO (90 min). $N = 3$ mice per group. Bar graphs represent the mean ± SD. ns, no significance. Unpaired Student's *t*-test in (A) and (B). Source data are available online for this figure.

