## [Peer Review File · EMBO Molecular Medicine]

Targeting HPK1 inhibits neutrophil responses to mitigate post-stroke lung and cerebral injuries

Xiao-Yu Hou, Tingting Zhang, Ying Sun, Jing Xia, Hong-Ye Fan, Dingfang Shi, Qian Wu, and Ming Huang

Corresponding author(s): Xiao-Yu Hou (xyhou@cpu.edu.cn) , Ming Huang (huangming88@xzhmu.edu.cn)

Review Timeline:

Submission Date:	25th Sep 24
Editorial Decision:	15th Oct 24
Revision Received:	4th Feb 25
Editorial Decision:	20th Feb 25
Revision Received:	1st Mar 25
Accepted:	10th Mar 25

Editor: Lise Roth

Transaction Report:

15th Oct 2024

Dear Prof. Hou,

Thank you for the submission of your manuscript to EMBO Molecular Medicine. We have now received feedback from the three reviewers who agreed to evaluate your manuscript. As you will see from the reports below, the referees acknowledge the interest of the study and are overall supporting publication of your work pending appropriate revisions.

Addressing the reviewers' concerns in full will be necessary for further considering the manuscript in our journal, and acceptance of the manuscript will entail a second round of review.

Please note that upon further discussion with the referees, we agreed that conditional knock-out experiments, additional human data, and blood-brain barrier crossing experiments will NOT be requested for further consideration of the manuscript, and these concerns could instead be addressed by adequate discussion and toning-down some of the conclusions.

As mentioned by referee #2 in particular, it will be important to streamline the manuscript to make the main message(s) accessible to the reviewers and readers.

EMBO Molecular Medicine encourages a single round of revision only and therefore, acceptance or rejection of the manuscript will depend on the completeness of your responses included in the next, final version of the manuscript. For this reason, and to save you from any frustrations in the end, I would strongly advise against returning an incomplete revision.

We are expecting your revised manuscript within three months, if you anticipate any delay, please contact us.

We require:

4) A .docx formatted letter INCLUDING the reviewers' reports and your detailed point-by-point responses to their comments. As part of the EMBO Press transparent editorial process, the point-by-point response is part of the Review Process File (RPF), which will be published alongside your paper.

5) A complete author checklist, which you can download from our author guidelines (<https://www.embopress.org/page/journal/17574684/authorguide#submissionofrevisions>). Please insert information in the checklist that is also reflected in the manuscript. The completed author checklist will also be part of the RPF.

6) All Materials and Methods need to be described in the main text using our 'Structured Methods' format. According to this format, the Methods section includes a Reagents and Tools Table (listing key reagents, experimental models, software and relevant equipment and including their sources and relevant identifiers) followed by a Methods and Protocols section describing the methods, ideally using a step-by-step protocol format. The aim is to facilitate adoption of the methodologies across labs. Please download and fill our Reagents and Tools Table template (.docx), which you can find in our author guidelines: <https://www.embopress.org/page/journal/14693178/authorguide#structuredmethods>.

<https://www.embopress.org/doi/10.15252/msb.20178071>

7) Please note that all corresponding authors are required to supply an ORCID ID for their name upon submission of a revised manuscript.

8) It is mandatory to include a 'Data Availability' section after the Materials and Methods. Before submitting your revision, primary datasets produced in this study need to be deposited in an appropriate public database, and the accession numbers and database listed under 'Data Availability'. Please remember to provide a reviewer password if the datasets are not yet public (see <https://www.embopress.org/page/journal/17574684/authorguide#dataavailability>).

9) For data quantification: please specify the name of the statistical test used to generate error bars and P values, the number (n) of independent experiments (specify technical or biological replicates) underlying each data point and the test used to calculate p-values in each figure legend. The figure legends should contain a basic description of n, P and the test applied. Graphs must include a description of the bars and the error bars (s.d., s.e.m.). Please provide exact p values.

10) Our journal encourages inclusion of *data citations in the reference list* to directly cite datasets that were re-used and obtained from public databases. Data citations in the article text are distinct from normal bibliographical citations and should directly link to the database records from which the data can be accessed. In the main text, data citations are formatted as follows: "Data ref: Smith et al, 2001" or "Data ref: NCBI Sequence Read Archive PRJNA342805, 2017". In the Reference list, data citations must be labeled with "[DATASET]". A data reference must provide the database name, accession number/identifiers and a resolvable link to the landing page from which the data can be accessed at the end of the reference. Further instructions are available at .

11) We replaced Supplementary Information with Expanded View (EV) Figures and Tables that are collapsible/expandable online. A maximum of 5 EV Figures can be typeset. EV Figures should be cited as 'Figure EV1, Figure EV2" etc... in the text and their respective legends should be included in the main text after the legends of regular figures.

12) The paper explained: EMBO Molecular Medicine articles are accompanied by a summary of the articles to emphasize the major findings in the paper and their medical implications for the non-specialist reader. Please provide a draft summary of your article highlighting

- the medical issue you are addressing,

- the results obtained and

- their clinical impact.

13) Author contributions: CRedit has replaced the traditional author contributions section because it offers a systematic machine readable author contributions format that allows for more effective research assessment. Please remove the Authors Contributions from the manuscript and use the free text boxes beneath each contributing author's name in our system to add specific details on the author's contribution. More information is available in our guide to authors.

Please also suggest a visual abstract to illustrate your article as a PNG file 550 px wide x 300-600 px high. A cropped portion of this image will serve as thumbnail for the table of content on our webpage.

16) As part of the EMBO Publications transparent editorial process initiative (see our Editorial at <http://embomolmed.embopress.org/content/2/9/329>), EMBO Molecular Medicine will publish online a Review Process File (RPF)

to accompany accepted manuscripts.

In the event of acceptance, this file will be published in conjunction with your paper and will include the anonymous referee reports, your point-by-point response and all pertinent correspondence relating to the manuscript. Let us know whether you agree with the publication of the RPF and as here, if you want to remove or not any figures from it prior to publication. Please note that the Authors checklist will be published at the end of the RPF.

I look forward to receiving your revised manuscript.

Yours sincerely,

Lise Roth

***** Reviewer's comments *****

Referee #1 (Comments on Novelty/Model System for Author):

Appropriate experiments are selected to demonstrate the effects of HPK1 and its loss/inhibition on inflammation-mediated outcomes in experimental stroke in mice. Although effects of HPK1 have been demonstrated in experimental inflammation, the authors here apply that knowledge in clinically relevant experimental models of disease.

Referee #1 (Remarks for Author):

Thanks for sending this manuscript through for review. I have some comments that I hope will prove useful for the authors.

1. The abstract makes reference to neutrophils being responsible for poor neurological outcomes after stroke. It may also be worth mentioning in the abstract that they have been implicated in respiratory morbidity after stroke in order to justify the scope of the study, which makes an assessment of (presumably neutrophil-mediated) post-stroke lung injury.
2. Line 53: "numbers of neutrophils rise in the first few hours [post-stroke]" - is it worth citing a primary source to support this assertion?
3. Line 66: it might be worth changing the emphasis of these lines so that they primarily talk about HPK1 function in neutrophils. I'm not certain its effects in T cells are relevant to this study.
4. I presume Fig 1B and 1C are dose-ranging studies using neutrophils from WT mice? I think it's worth making that clear in the text (around line 90) and the figure, as well as in the figure caption.
5. Line 100: "... did not induce changes in neutrophil activity in physiological conditions..." I don't think the authors can make that assertion given that what is demonstrated in the figure are assessments of mRNA expression rather than functional neutrophil assays. I would suggest revision so that the conclusion is supported by the data presented.
6. The methods make no mention of how flow cytometry gates were set. I think this is a significant omission given the appearance of different positions of the neutrophil gate in figure 5B between sham and MCAO groups. The authors should make it clear how the gates were set (using fluorescence-minus-one controls, maybe?) and given the apparently different positions of the gate shown in fig 5B, I think there should be evidence presented (perhaps in supplementary data) that the gates were set in

at least a semi-objective fashion.

7. Line 137: I think the authors can be more specific that the assay is a measure of alveolar capillary barrier disruption.

8. Fig 4C: was an objective measure (perhaps MFI) used to quantify CitH3 immunofluorescence in WT vs KO cortex following MCAO/R? Can the authors provide these data as well as the IF image?

9. Line 148: the authors assert that "HPK1-induced neutrophil infiltration leads to inflammatory responses and neutrophil NETosis", but the data provided do not support that the infiltration itself is induced by HPK1 (merely mediated by it at least in part), or that NETosis is a necessary consequence thereof. I would suggest rewording this sentence for it to summarise the data provided more accurately.

10. Fig 5A: I would suggest making it clearer that these data are from WT mice in the text and in the figure.

11. Line 241: the authors assert that circulating neutrophils are "responsible for stroke severity and functional outcome", which may be too strong an assertion for the available evidence. It might be worth softening this to "neutrophil responses contribute to" or something similar.

12. Line 269: here and later, I think NF-kappaB, STAT3 and p38 pathways should be considered and talked about separately given the way they are tested in the experiments (despite, I appreciate, there being some overlap in pathways after activation).

13. Line 291: the Moriya study was a phase 1 study and almost certainly underpowered to detect significant differences in change in Barthel index, so I do not think the citation supports the assertion that G-CSF results in a post-stroke function improvement.

14. Line 299: the Lazaar study was a phase 2b study and could not draw any conclusions about the clinical efficacy of the CXCR2 inhibitor in COPD, so the citation does not support the assertion. Similarly the Todd study was able to show effects on in-vitro migration of neutrophils from peripheral blood, but cannot be used to support the assertion that the inhibitor "alleviates" atopic asthma.

15. Line 302: I think the claim that HPK1 drives PMN CXCR2 expression is probably too strong an assertion for the evidence provided by this study. I would suggest rewording this.

16. Line 338: the authors assert that the HPK1 inhibitor they mention has a "broad therapeutic window", citing the Deva abstract. The Deva abstract can only show safety at most in the small group of people in the phase 1a study, rather than demonstrate a therapeutic window. I would soften this claim.

17. I think the discussion should place a much greater emphasis on potential limitations of the study.

18. The authors suggest future research involving neutrophil-specific knockout of HPK1: I am not certain that is a useful next line of inquiry given that - if we take the data as presented - the authors demonstrate what would be clinical efficacy with systemic HPK1 inhibition. I think the authors should consider more carefully what future research might follow from this.

Thanks again for sending through this manuscript for review.

Referee #2 (Comments on Novelty/Model System for Author):

The paper is an exhaustive study of the role of HPK1 in neutrophil mobilization covering several models and using in vitro and in vivo resources.

Referee #2 (Remarks for Author):

The paper by Zhang and colleagues investigates the role of HPK1 in neutrophil mobilisation and activation. Whilst the paper is clearly written and the results extremely comprehensive it might be useful for the authors to realise that sometimes 'less is more'. At the start of the paper, the manuscript reads as if we are going to study neutrophil mobilisation to the brain and lung post-stroke, we then move onto the mechanisms of HPK1 neutrophil activation in vitro, then we jump right out into a lung-injury model. Part of the art of writing a paper involves selecting the data which tell the story you're interested in telling, not necessarily putting all the data you've ever collected in one big list. Given the slightly overwhelming amount of data presented, I have the following queries:

1. The authors only used male mice in this study. Given the differences in inflammatory response in males and females, and the enormity of this study, can they justify their use of only one sex in the discussion?

2. Figure 4C is unclear. The TTC staining shows a large area of the cortex in the WT animals is dead, can the authors show which area of the cortex they analysed for neutrophil infiltration using some kind of schematic?

This is where we hit a bit of a snag. Figure 7 is where we hit a problem. Figures 1-6 are actually fine, if a little numerous (do the FACS plots need to be in the main text - could they go supplementary?). The story seems to go 'HPK1 affects neutrophil mobilisation to the brain and lung post-stroke, HPK1 affects neutrophil activation, HPK1 affects neutrophil activation by this mechanism (NFkB/STAT stuff - I would also query the need for all this mechanistic stuff AND the CXCR2 stuff but that's for you to decide).' But what happens at the end is that we've switched from KO animals to an inhibitor. Which is fine, but there's not a great deal of continuity.

3. Why not do the mechanistic studies of NFkB/STAT/etc. in neutrophils from the KO animals?

4. Why do the authors not look at lung injury after MCAO in the i-HPK animals? This would reflect the work in the initial figures and confirm that HPK1 is involved in neutrophil migration to the lungs and 'bookend' the paper well.

5. What is the contribution of the lung injury model? I mean, I see why the authors have done it, from a scientific point of view it's important to understand whether this mechanism is consistent across different injuries. But if that's your aim then why not also focus on why neutrophils ARE going to the spleen in both animals? I think if the authors' main focus is (as their title suggests) neutrophils, HPK1 and stroke-induced brain and lung injury then they need to stick to stroke-induced brain and lung injury and lose the random model in the middle. It can easily be moved to supplementary. Less is more!

Otherwise the only issue I had with the manuscript is that the figure labelling is sometimes a little confusing. For example figure 2C is three graphs but placed vertically, figure 4B is two graphs but horizontally. Why not label all the individual graphs? If the flow cytometry plots were moved to the supplementary this would give you considerably more space to do this, it almost feels like you were worried you would run out of letters!

The discussion, introduction and methods are well written and comprehensive and I have no major issues with them.

Referee #3 (Remarks for Author):

This study explores the role of hematopoietic progenitor kinase 1 (HPK1), also known as mitogen-activated protein kinase kinase kinase 1 (MAP4K1), in regulating neutrophil responses following acute ischemic stroke (AIS). The authors have shown that HPK1 drives neutrophil hyperactivation via CXCR2, NF- κ B/STAT3/p38 pathways, and Gasdermin D cleavage, leading to systemic inflammation and tissue damage. Loss or inhibition of HPK1 reduced neutrophil activity, alleviating lung and neurological impairments in mice after AIS. They claim that the findings identify HPK1 as a key regulator of neutrophil-driven inflammation and a potential therapeutic target for improving outcomes post-stroke. I have the following comments on the study.

HPK1-induced cell activation and neutrophil recruitment have already been reported. This would reduce the novelty of the study. In addition, they evaluated NFkB and GSDMD pathways, both of which have already been studied extensively. Several studies have previously elucidated the role of MAP4K1 in inflammation, yet the mechanism outlined in this manuscript lacks precision. The authors extend the pathway from NF- κ B to Gasdermin D. However, it is well-established that while NF- κ B can be activated by a single stimulus, optimal activation of the Gasdermin D-mediated pathway requires two signals-NF- κ B and inflammasome activation. The authors should provide a clearer explanation of how HPK1 regulates the cross-talk between these two pathways.

To accurately characterize acute lung injury (ALI), the authors should adhere to the ATS criteria, which require confirmation of at least three specific parameters to diagnose ALI (Am J Respir Cell Mol Biol. 2022 Feb; 66(2): e1-e14; PMC8845128).

The manuscript lacks evidence regarding the ability of the pharmacological inhibitor of HPK1 to cross the blood-brain barrier, which is a crucial consideration. The authors should present supporting data to validate this claim. To be more specific, can i-HPK1 cross the blood brain barrier, or does it affect on peripheral neutrophils subsequently migrating into the brain?

A significant limitation of this study is the absence of conditional knockout mice (neutrophil specific HPK1). The authors should address whether the observed effects in HPK1 knockout mice could be influenced by gender-specific effects or the activity of other immune cells, particularly in the context of AIS. Furthermore, including human data would significantly strengthen the manuscript's findings.

Do the authors have any data showing neutrophil HPK1 is upregulated in AIS?

Fig 1. Do the high doses of LPS really reflect the physiological condition of AIS? Those doses are used for studying sepsis, which is accompanied by severe endotoxemia. According to the serum cytokine levels in Figure 2C, the circulating LPS levels do not seem so high in the present AIS model.

HPK1 upregulation in neutrophils was demonstrated by 100 ng/mL LPS stimulation (Fig 1C), whereas NETs were induced by 10 μ g/mL LPS (Fig 1F). Did the authors confirm HPK1 upregulation by 10 μ g/mL LPS? HPK1 peaked at 3 h and decreased at 6 h. Is there any explanation on this phenomenon?

Fig 2B, C. The authors mentioned the "number" of neutrophils, but the data only shows the frequency, which could be affected by changes in other cell populations. Blood neutrophil count is usually presented in number/ μ L. In the bone marrow and other tissues, the number can be calculated using the counting beads.

Fig 2C-E. Sham group is missing. TNF- α and IL-1 β were in the very low range of the kit standard curve according to the

datasheet. Were these significantly higher than the sham?

Fig 2E. NETs were induced by LPS in neutrophils isolated from AIS mice. Why LPS was needed if neutrophils were already activated in AIS? How does this experiment correlate with the physiological setting?

Fig 3. Again, sham group is missing. Fig 3A and 4A only show the frequency not the number.

Fig 3B, 4B. For tissue cytokine levels, it would make more sense to be presented per weight or protein amount rather than volume.

Fig 5A-B. It would be better to provide the cell number in addition to frequency.

Fig 5C and 5E as well as 5D and 5F can be done in the same experiment and presented in the same graph.

Fig 6. Are these relevant in vitro experiments for AIS?

Fig 7. Why did the authors choose different time points from Fig 3 for some experiments? The same thing can be applied between Fig 4 and Fig 8.

Fig 7D. Why is the % of CitH3 positive cells in the vehicle group so different from Fig 3C WT group? The time points are different, but why did the authors choose to do so?

Minor:

Fig 3B. The legend should be lung instead of brain.

Responses to the Reviewers' comments

Manuscript number: EMM-2024-20638-V2

Title: Targeting HPK1 inhibits neutrophil responses to mitigate post-stroke lung and cerebral injuries

Authors: Tingting Zhang, Ying Sun, Jing Xia, Hongye Fan, Dingfang Shi, Qian Wu, Ming Huang, Xiao-Yu Hou

Response to Referee #1

Comments on Novelty/Model System for Author:

Appropriate experiments are selected to demonstrate the effects of HPK1 and its loss/inhibition on inflammation-mediated outcomes in experimental stroke in mice. Although effects of HPK1 have been demonstrated in experimental inflammation, the authors here apply that knowledge in clinically relevant experimental models of disease.

Remarks for Author:

Thanks for sending this manuscript through for review. I have some comments that I hope will prove useful for the authors.

Comment 1: The abstract makes reference to neutrophils being responsible for poor neurological outcomes after stroke. It may also be worth mentioning in the abstract that they have been implicated in respiratory morbidity after stroke in order to justify the scope of the study, which makes an assessment of (presumably neutrophil-mediated) post-stroke lung injury

Response: We greatly appreciate your valuable advice and positive comments on our study. Per your recommendation, we have revised the Abstract to include the statement that circulating neutrophils have been implicated in respiratory morbidity after acute ischemic stroke (AIS). (**Lines 27-28**).

Comment 2: Line 53: "numbers of neutrophils rise in the first few hours [post-stroke]" - is it worth citing a primary source to support this assertion?

Response: Thank you for your valuable suggestion. As you suggested, we have cited the articles (Ross *et al*, 2007; Jickling *et al*, 2015 [PMID:17845917 and PMID: 25806703]) to support this assertion in the revised Introduction. (**Lines 58-60**).

Comment 3: Line 66: it might be worth changing the emphasis of these lines so that they primarily talk about HPK1 function in neutrophils. I'm not certain its effects in T cells are relevant to this study.

Response: Thank you for your insightful comments. We agree that HPK1 effects in T cells are not directly relevant to this study. Based on your comment, we have revised these lines to emphasize on HPK1 function in neutrophils during acute inflammation in the revised Introduction section as follows:

“HPK1 promotes CXC chemokine ligand 1 (CXCL1)-induced neutrophil adhesion under flow conditions, probably through phosphorylating adaptor protein HIP-55 (Schymeinsky *et al*, 2009; Jakob *et al*, 2013; Chuang *et al*, 2016). Moreover, HPK1 positively regulates neutrophil adhesion and recruitment following tumor necrosis factor (TNF)- α -induced acute local inflammation in mouse cremaster muscle (Jakob *et al*, 2013).” **(Lines 80-85).**

Comment 4: I presume Fig 1B and 1C are dose-ranging studies using neutrophils from WT mice? I think it's worth making that clear in the text (around line 90) and the figure, as well as in the figure caption.

Response: Thank you for bringing this to our attention. As you presumed, Fig 1B and 1C show dose-ranging studies using neutrophils from WT mice. In the revised manuscript, we have indicated that primary neutrophils were derived from WT mice in the text **(Line 106)**, Fig 1D and Fig EV1, and figure legends.

Comment 5: Line 100: "... did not induce changes in neutrophil activity in physiological conditions..." I don't think the authors can make that assertion given that what is demonstrated in the figure are assessments of mRNA expression rather than functional neutrophil assays. I would suggest revision so that the conclusion is supported by the data presented.

Response: We apologize for making the confusion. To clarify this, we have revised the related results as follows:

“Under physiological conditions, HPK1-KO neutrophils exhibited levels of proinflammatory factor transcripts **(Fig 1E-H)**, ROS **(Fig 1I-J)**, and NETs **(Fig 1K-L)** comparable to those of WT controls.” **(Lines 114-117).**

Comment 6: The methods make no mention of how flow cytometry gates were set. I think this is a significant omission given the appearance of different positions of the neutrophil gate in figure 5B between sham and MCAO groups. The authors should make it clear how the gates were set (using fluorescence-minus-one controls, maybe?) and given the apparently different positions of the gate shown in fig 5B, I think there should be evidence presented (perhaps in supplementary data) that the gates were set in at least a semi-objective fashion.

Response: Thank you for this valuable suggestion. We have added the settings of the

flow cytometry gates to the revised Methods section as follows:

“The gating strategy for determining neutrophil population in the blood, bone marrow, lung, and spleen was set as follows: First, debris was excluded from single cells. Next, leukocytes were gated as CD45⁺ cells. Finally, neutrophils were gated as CD11b⁺ and Ly6G⁺ cells from the CD45⁺ cells. The gating strategy for brain neutrophils and microglia was set as follows: microglia were gated as CD45^{intermediate} and CD11b⁺ cells, while neutrophils were gated as CD11b⁺ and Ly6G⁺ cells from the CD45^{high} population.” (Lines 450-457).

In addition, compensation controls were used to minimize the overlap of different fluorescent channels to set the gates. We have submitted the source data of the compensation controls. The apparently different positions in previous edition of Fig. 5B may result from batch effects. In the revised manuscript, the representative flow cytometry images in the context of same voltages have been shown.

Comment 7: Line 137: I think the authors can be more specific that the assay is a measure of alveolar capillary barrier disruption.

Response: Thank you for your pertinent suggestion. In the revised manuscript, we have revised the description of the assay to specifically state that it is a measure of alveolar capillary barrier disruption. (Lines 199-200).

Comment 8: Fig 4C: was an objective measure (perhaps MFI) used to quantify CitH3 immunofluorescence in WT vs KO cortex following MCAO/R? Can the authors provide these data as well as the IF image?

Response: Thank you for the valuable suggestion. Per your suggestion, we have supplemented the quantification of CitH3 immunofluorescence in the revised Fig 6I and Fig 8K.

Comment 9: Line 148: the authors assert that "HPK1-induced neutrophil infiltration leads to inflammatory responses and neutrophil NETosis", but the data provided do not support that the infiltration itself is induced by HPK1 (merely mediated by it at least in part), or that NETosis is a necessary consequence thereof. I would suggest rewording this sentence for it to summarise the data provided more accurately.

Response: Thank you for highlighting this important point. As you suggested, we have revised this sentence in the manuscript as follows:

“These results suggest that HPK1 mediates neutrophil infiltration, inflammatory responses, and neutrophil NETosis, at least in part, thereby aggravating acute lung injury and neurological deficits following ischemic

stroke.” (Lines 225-227).

Comment 10: Fig 5A: I would suggest making it clearer that these data are from WT mice in the text and in the figure.

Response: As you suggested, we have made it clear that the data are from WT mice in the revised text (Lines 147 and 171), in the revised Fig 3D-E and Fig 4B-C, and the corresponding legends.

Comment 11: Line 241: the authors assert that circulating neutrophils are "responsible for stroke severity and functional outcome", which may be too strong an assertion for the available evidence. It might be worth softening this to "neutrophil responses contribute to" or something similar.

Response: Thank you for your insightful comments. Per your suggestion, we have revised the sentence as follows: “Circulating neutrophil responses are associated with stroke severity and functional outcomes.” (Lines 270-271).

Comment 12: Line 269: here and later, I think NF-kappaB, STAT3 and p38 pathways should be considered and talked about separately given the way they are tested in the experiments (despite, I appreciate, there being some overlap in pathways after activation).

Response: Thank you very much for this valuable suggestion. As suggested, we have discussed the NF- κ B, STAT3 and p38 pathways separately in the revised manuscript. (Lines 290-307).

Comment 13: Line 291: the Moriya study was a phase 1 study and almost certainly underpowered to detect significant differences in change in Barthel index, so I do not think the citation supports the assertion that G-CSF results in a post-stroke function improvement.

Response: Thank you for noting this important point. We have deleted this citation in the revised manuscript.

Comment 14: Line 299: the Lazaar study was a phase 2b study and could not draw any conclusions about the clinical efficacy of the CXCR2 inhibitor in COPD, so the citation does not support the assertion. Similarly the Todd study was able to show effects on in-vitro migration of neutrophils from peripheral blood, but cannot be used to support the assertion that the inhibitor "alleviates" atopic asthma.

Response: Thank you for pointing this out. We have deleted these two citations in the

revised manuscript.

Comment 15: Line 302: I think the claim that HPK1 drives PMN CXCR2 expression is probably too strong an assertion for the evidence provided by this study. I would suggest rewording this.

Response: Thank you for your pertinent suggestion. We have revised this sentence as follows:

“Therefore, HPK1 is involved in CXCR2 expression, accounting for neutrophil mobilization into the blood and infiltration into the brain and lungs after stroke.” (Lines 319-321).

Comment 16: Line 338: the authors assert that the HPK1 inhibitor they mention has a "broad therapeutic window", citing the Deva abstract. The Deva abstract can only show safety at most in the small group of people in the phase 1a study, rather than demonstrate a therapeutic window. I would soften this claim.

Response: As you said, the Deva study was a phase 1a study that preliminarily evaluated safety. Therefore, we have revised the sentence as follows:

“i-HPK1 has demonstrated high selectivity and efficiency in the pre-clinical studies and proven safe in clinical studies (Deva et al, 2024; Liu et al, 2022).” (Lines 340-342).

Comment 17: I think the discussion should place a much greater emphasis on potential limitations of the study.

Response: Thank you for your careful observation. We have supplemented the limitations in the Discussion and last paragraph as follows:

“A significant limitation of this study is the absence of data from patients with AIS. Further studies obtaining patient data from peripheral blood will benefit i-HPK1 clinical translation. In addition, HPK1 is expressed in macrophages and lymphocytes. Further studies are required to investigate HPK1 roles in specific leukocytes following AIS. Behavioral experiments are also essential to evaluate the role of HPK1 in long-term outcomes.” (Lines 345-350).

Comment 18: The authors suggest future research involving neutrophil-specific knockout of HPK1: I am not certain that is a useful next line of inquiry given that - if we take the data as presented - the authors demonstrate what would be clinical efficacy with systemic HPK1 inhibition. I think the authors should consider more carefully what future research might follow from this.

Response: We agree with the suggestion. In combination with the Editor's and Reviewers' comments, we have revised the limitations in the last paragraph of the Discussion. **(Lines 345-350).**

Manuscript number: EMM-2024-20638-V2

Title: Targeting HPK1 inhibits neutrophil responses to mitigate post-stroke lung and cerebral injuries

Authors: Tingting Zhang, Ying Sun, Jing Xia, Hongye Fan, Dingfang Shi, Qian Wu, Ming Huang, Xiao-Yu Hou

Response to Referee #2

Comments on Novelty/Model System for Author:

The paper is an exhaustive study of the role of HPK1 in neutrophil mobilization covering several models and using *in vitro* and *in vivo* resources.

Remarks for Author:

The paper by Zhang and colleagues investigates the role of HPK1 in neutrophil mobilisation and activation. Whilst the paper is clearly written and the results extremely comprehensive it might be useful for the authors to realise that sometimes 'less is more'. At the start of the paper, the manuscript reads as if we are going to study neutrophil mobilisation to the brain and lung post-stroke, we then move onto the mechanisms of HPK1 neutrophil activation *in vitro*, then we jump right out into a lung-injury model. Part of the art of writing a paper involves selecting the data which tell the story you're interested in telling, not necessarily putting all the data you've ever collected in one big list. Given the slightly overwhelming amount of data presented, I have the following queries:

Response: We appreciate your positive and constructive comments and valuable suggestions for improving our study. Following your suggestion, we have streamlined the manuscript to make the main messages accessible to the readers. First, we investigated the involvement of HPK1 in neutrophil responses *in vitro*. Next, we assessed the effects of HPK1 on systemic inflammation and neutrophil mobilization in a mouse model of ischemic stroke. Finally, both genetic deletion and pharmacological inhibition highlighted HPK1 as a novel therapeutic target for alleviating post-stroke lung and brain injuries.

Comment 1: The authors only used male mice in this study. Given the differences in inflammatory response in males and females, and the enormity of this study, can they justify their use of only one sex in the discussion?

Response: Thank you for this valuable suggestion. In response to the suggestion of Referee #3, we have performed additional studies to determine whether the observed effects of HPK1 on pulmonary and cerebral injuries could be influenced by sex-specific effects in the context of AIS. The results are presented in the Results section and Fig EV3 in the revised manuscript (**Lines 215-224**). Briefly, similar to its effects in male mice, HPK1 deficiency alleviated acute lung injury and neurological

deficits after ischemic stroke in female mice.

Comment 2: Figure 4C is unclear. The TTC staining shows a large area of the cortex in the WT animals is dead, can the authors show which area of the cortex they analysed for neutrophil infiltration using some kind of schematic?

This is where we hit a bit of a snag. Figure 7 is where we hit a problem. Figures 1-6 are actually fine, if a little numerous (do the FACS plots need to be in the main text - could they go supplementary?). The story seems to go 'HPK1 affects neutrophil mobilisation to the brain and lung post-stroke, HPK1 affects neutrophil activation, HPK1 affects neutrophil activation by this mechanism (NFkB/STAT stuff - I would also query the need for all this mechanistic stuff AND the CXCR2 stuff but that's for you to decide).' But what happens at the end is that we've switched from KO animals to an inhibitor. Which is fine, but there's not a great deal of continuity.

Response: Thank you for your valuable comments and suggestions. As suggested, we have added schematic figures to show the neutrophil infiltration area in the revised Fig 6H and Fig 8J.

In addition, we removed the representative FACS plots and listed the gating strategies in the Figures and Methods. Following your suggestion on continuity, we have streamlined the manuscript to make the main messages accessible to the readers. First, we investigated the involvement of HPK1 in neutrophil responses *in vitro*. Next, we assessed the effects of HPK1 on systemic inflammation and neutrophil mobilization in a mouse model of ischemic stroke. Finally, both genetic deletion and pharmacological inhibition highlighted HPK1 as a novel therapeutic target for alleviating post-stroke lung and brain injuries.

Comment 3: Why not do the mechanistic studies of NFkB/STAT/etc. in neutrophils from the KO animals?

Response: Thank you for your pertinent suggestion. We have supplemented the mechanistic studies of NF-kB/STAT/p38-MAPK signaling pathway and GSDMD cleavage in neutrophils from the KO mice and accordingly added the data to the Results (**Lines 119-138**) and Fig 2A-H in the revised manuscript. These results suggest that neutrophil-intrinsic HPK1 aggravates inflammatory responses and NETosis by promoting NF-κB/STAT3/p38-MAPK and GSDMD pathway activation.

Comment 4: Why do the authors not look at lung injury after MCAO in the i-HPK animals? This would reflect the work in the initial figures and confirm that HPK1 is involved in neutrophil migration to the lungs and 'bookend' the paper well.

Response: Actually, we studied lung injury after MCAO in the i-HPK animals in the previous version of the manuscript, which is also shown in the Results (**Lines**

240-247) and Fig 7A-G in the revised manuscript. Based on your suggestion, we deleted the results using the lung injury model.

Comment 5: What is the contribution of the lung injury model? I mean, I see why the authors have done it, from a scientific point of view it's important to understand whether this mechanism is consistent across different injuries. But if that's your aim then why not also focus on why neutrophils ARE going to the spleen in both animals? I think if they authors' main focus is (as their title suggests) neutrophils, HPK1 and stroke-induced brain and lung injury then they need to stick to stroke-induced brain and lung injury and lose the random model in the middle. It can easily be moved to supplementary. Less is more!

Otherwise, the only issue I had with the manuscript is that the figure labelling is sometimes a little confusing. For example, figure 2C is three graphs but placed vertically, figure 4B is two graphs but horizontally. Why not label all the individual graphs? If the flow cytometry plots were moved to the supplementary this would give you considerably more space to do this, it almost feels like you were worried you would run out of letters!

Response: Thank you for the constructive and valuable suggestion. We strongly agreed that “Less is more”. Following your suggestion, we have deleted the results from the LPS-induced lung-injury model.

In addition, we have re-labeled all the panels in the revised Figures. The gating strategies of the flow cytometry analysis were shown in the revised Figures and Methods and the flow cytometry plots were submitted in source data.

Manuscript number: EMM-2024-20638-V2

Title: Targeting HPK1 inhibits neutrophil responses to mitigate post-stroke lung and cerebral injuries

Authors: Tingting Zhang, Ying Sun, Jing Xia, Hongye Fan, Dingfang Shi, Qian Wu, Ming Huang, Xiao-Yu Hou

Response to Referee #3

Remarks for Author:

This study explores the role of hematopoietic progenitor kinase 1 (HPK1), also known as mitogen-activated protein kinase kinase kinase 1 (MAP4K1), in regulating neutrophil responses following acute ischemic stroke (AIS). The authors have shown that HPK1 drives neutrophil hyperactivation via CXCR2, NF- κ B/STAT3/p38 pathways, and Gasdermin D cleavage, leading to systemic inflammation and tissue damage. Loss or inhibition of HPK1 reduced neutrophil activity, alleviating lung and neurological impairments in mice after AIS. They claim that the findings identify HPK1 as a key regulator of neutrophil-driven inflammation and a potential therapeutic target for improving outcomes post-stroke. I have the following comments on the study.

Comment 1: HPK1-induced cell activation and neutrophil recruitment have already been reported. This would reduce the novelty of the study. In addition, they evaluated NF κ B and GSDMD pathways, both of which have already been studied extensively. Several studies have previously elucidated the role of MAP4K1 in inflammation, yet the mechanism outlined in this manuscript lacks precision. The authors extend the pathway from NF- κ B to Gasdermin D. However, it is well-established that while NF- κ B can be activated by a single stimulus, optimal activation of the Gasdermin D-mediated pathway requires two signals-NF- κ B and inflammasome activation. The authors should provide a clearer explanation of how HPK1 regulates the cross-talk between these two pathways.

Response: Thank you for your valuable comments. We apologize for the confusion. Compared to its roles in adaptive immunity and tumor immunosuppression (Chuang *et al*, 2016; Hernandez *et al*, 2018; Si *et al*, 2020; Sun *et al*, 2023), the functional impact of HPK1 on innate immunity has been ignored. Previous study has shown that HPK1 positively regulates neutrophil adhesion after acute inflammation (Jakob *et al*, 2013). This study expands on the understanding that HPK1 plays a critical role in innate immune regulation in an AIS model. We found that HPK1 mediates post-stroke neutrophil mobilization, systemic inflammation, as well as and various neutrophil responses, involving NF- κ B/STAT3/p38 MAPK and GSDMD signaling pathways. This study highlights HPK1 as a promising therapeutic target for AIS therapy.

We have further provided an explanation of how HPK1 regulates the cross-talk between these two pathways in the revised Discussion (**Lines 290-297**).

Comment 2: To accurately acute lung injury (ALI), the authors should adhere to the ATS criteria, which require confirmation of at least three specific parameters to diagnose ALI (Am J Respir Cell Mol Biol. 2022 Feb; 66(2): e1-e14; PMC8845128).

Response: According to the ATS criteria, lung injury is characterized by four parameters: histological evidence of tissue, alteration of the alveolar-capillary barrier, presence of an inflammatory response, and evidence of physiological dysfunction. As you suggested, we adhered to the ATS criteria by confirming the following three parameters used to diagnose ALI:

1. HE staining: histological evidence of tissue.
2. BALF concentration: alteration of the alveolar-capillary barrier.
3. Cytokine levels and/or neutrophil infiltration: presence of an inflammatory response.

Comment 3: The manuscript lacks evidence regarding the ability of the pharmacological inhibitor of HPK1 to cross the blood-brain barrier, which is a crucial consideration. The authors should present supporting data to validate this claim. To be more specific, can i-HPK1 cross the blood brain barrier, or does it affect on peripheral neutrophils subsequently migrating into the brain?

Response: Thank you for the comment. The data in Fig 7D and 7E showed that i-HPK1 inhibits neutrophil infiltration in the lungs after MCAO/R for 24 h, suggesting that i-HPK1 rapidly affects peripheral neutrophils trafficking. Moreover, the data in Fig 8E-F and 8I-K showed that i-HPK1 inhibits neutrophil infiltration and NET formation in the ischemic hemisphere at the early reperfusion stage (MCAO/R for 24 hours). Peripheral neutrophil responses and inflammation enhance the BBB permeability, allowing i-HPK1 cross the BBB into the brain after AIS. HPK1 is not expressed in brain cells. Therefore, it would be interesting to investigate the roles of HPK1 in distinct leukocytes. We have added this limitation to the revised Discussion (Lines 347-349).

Comment 4: A significant limitation of this study is the absence of conditional knockout mice (neutrophil specific HPK1). The authors should address whether the observed effects in HPK1 knockout mice could be influenced by gender-specific effects or the activity of other immune cells, particularly in the context of AIS. Furthermore, including human data would significantly strengthen the manuscript's findings.

Response: Thank you for this valuable suggestion. Per your suggestion, we have performed additional studies to determine whether the observed effects of HPK1 on pulmonary and cerebral injuries could be influenced by sex-specific effects in the

context of AIS. The results are presented in the Results (**Lines 215-224**) and Fig EV3. Briefly, similar to its effects in male mice, HPK1 knockout alleviated acute lung injury and neurological deficits after ischemic stroke in female mice.

We agree that human data would significantly strengthen the manuscript's findings. Additionally, it would be interesting to investigate the roles of HPK1 in distinct leukocytes after ischemic stroke. We have accordingly added these limitations in the revised Discussion (**Lines 345-349**).

Comment 5: Do the authors have any data showing neutrophil HPK1 is upregulated in AIS?

Response: Thank you for the valuable comment. In the revised manuscript, we have added data indicating that HPK1 was upregulated in circulating neutrophils from WT mice after MCAO/R for 6 h (Fig 3A and 3B).

Comment 6: Fig 1. Do the high doses of LPS really reflect the physiological condition of AIS? Those doses are used for studying sepsis, which is accompanied by severe endotoxemia. According to the serum cytokine levels in Figure 2C, the circulating LPS levels do not seem so high in the present AIS model.

Response: Thank you for this important observation. After an ischemic stroke, early neuronal cell death results in the release of DAMPs, which acts on the TLRs of leukocytes to induce the release of various pro-inflammatory cytokines into the bloodstream. As LPS is an activator of TLR4, we applied LPS stimulation *in vitro* to mimic the impact of the post-stroke inflammatory state on neutrophils.

Comment 7: HPK1 upregulation in neutrophils was demonstrated by 100 ng/mL LPS stimulation (Fig 1C), whereas NETs were induced by 10 µg/mL LPS (Fig 1F). Did the authors confirm HPK1 upregulation by 10 µg/mL LPS? HPK1 peaked at 3 h and decreased at 6 h. Is there any explanation on this phenomenon?

Response: Thank you for the comment. We have confirmed HPK1 upregulation by 10 µg/mL LPS and the data were shown in Fig EV1C and 1D.

As shown in the revised Fig EV1A and 1B, HPK1 expression was consistently upregulate. Although the levels of HPK1 after 6 h of LPS treatment seemed to produce a slight decline compared to those after LPS treatment for 3 h, there was no statistically significant difference between these time points.

Comment 8: Fig 2B, C. The authors mentioned the "number" of neutrophils, but the data only shows the frequency, which could be affected by changes in other cell populations. Blood neutrophil count is usually presented in number/µL. In the bone

marrow and other tissues, the number can be calculated using the counting beads.

Response: Thank you for pointing this out. According to your suggestion, we have added neutrophil count (numbers/ μ L) in the revised manuscript. In the bone marrow and other tissues, the neutrophil count was presented as numbers/tissue.

Comment 9: Fig 2C-E. Sham group is missing. TNF- α and IL-1 β were in the very low range of the kit standard curve according to the datasheet. Were these significantly higher than the sham?

Response: As you suggested, we have added the respective sham groups to the revised Fig 3H-J. TNF- α and IL-1 β levels from the MCAO group in WT mice were significantly higher than those from the sham group.

Comment 10: Fig 2E. NETs were induced by LPS in neutrophils isolated from AIS mice. Why LPS was needed if neutrophils were already activated in AIS? How does this experiment correlate with the physiological setting?

Response: Thank you for your pertinent observation. NETs are extracellular web-like structures composed of neutrophil-derived DNA, histones, and granzymes. After AIS, NETs are released into the blood by hyperactivated neutrophils through a process known as NETosis. In this study, we isolated circulating neutrophils using the Histopaque 1077/1119 method to obtain cells with intact cellular structure. To determine the ability of activated neutrophils to form NETs after AIS, LPS was used to stimulate neutrophils to form NETs. Our data suggest that neutrophils isolated after ischemic stroke are hyperactivated and have a stronger ability to form NETs than those isolated from the sham group.

Comment 11: Fig 3. Again, sham group is missing. Fig 3A and 4A only show the frequency not the number.

Response: According to your suggestion, we have added the respective sham groups to the revised Fig 5, and the neutrophil numbers to the revised Fig 5B and Fig 6B.

Comment 12: Fig 3B, 4B. For tissue cytokine levels, it would make more sense to be presented per weight or protein amount rather than volume.

Response: We apologize for these incorrected labels. In the revised manuscript, we have corrected those labels according to the protein amount (pg/mg).

Comment 13: Fig 5A-B. It would be better to provide the cell number in addition to frequency

Response: As suggested, we have provided neutrophil counts in the revised manuscript.

Comment 14: Fig 5C and 5E as well as 5D and 5F can be done in the same experiment and presented in the same graph.

Response: Thank you for the comment. As suggested, the data have been presented in the same graph (Fig 4F-G, 4H-I, and 4J-K) in the revised manuscript.

Comment 15: Fig 6. Are these relevant in vitro experiments for AIS?

Response: These experiments were performed to determine the effect of the HPK1 inhibitor on neutrophil responses. Considering continuity, we have moved Fig 6 to Fig EV4 in the revised manuscript.

Comment 16: Fig 7. Why did the authors choose different time points from Fig 3 for some experiments? The same thing can be applied between Fig 4 and Fig 8.

Response: After ischemic stroke, an inflammatory response occurs early, contributing to tissue injury later. Therefore, we observed neutrophil infiltration and cytokines levels after MCAO/R for 24 h and observed brain and lung injuries after MCAO/R for 48 or 72 h.

Comment 17: Fig 7D. Why is the % of CitH3 positive cells in the vehicle group so different from Fig 3C WT group? The time points are different, but why did the authors choose to do so?

Response: Thank you for pointing this out. Considering that acute lung injury is correlated with stroke severity, a severe ischemia (90-min occlusion) was performed to evaluate the therapeutic potential of i-HPK1 in ameliorating post-stroke outcomes.

Comment 18: Minor: Fig 3B. The legend should be lung instead of brain.

Response: Thank you for pointing this out. In the revised manuscript, we have corrected the legends and thoroughly checked the text and figures.

20th Feb 2025

Dear Prof. Hou,

Thank you for submitting your revised study. We have now received the reports from the three referees who evaluated your revised manuscript. As you will see from the reports below, they are overall satisfied with the revisions, and I will therefore be able to accept your manuscript once the following editorial issues are addressed:

1/ Referee 2#'s comments:

- Regarding the re-organization of figures, please note that you have the possibility to have more EV figures if you wish to remove some panels from the main figures.
- Please consider the comment on inclusion of female animals.

2/ Manuscript text:

- Please remove the yellow highlights and only keep in track changes mode any new modification.
- We can accommodate a maximum of 5 keywords, please adjust accordingly.
- Methods:
 - o Please remove the reagent and tools table from the manuscript.
 - o Cells: please indicate whether the cells were tested for mycoplasma contamination.
 - o Statistical analysis: please provide a statement on sample size, randomization and blinding.
- Author contributions: CRediT has replaced the traditional author contributions section because it offers a systematic machine readable author contributions format that allows for more effective research assessment. Please remove the Authors Contributions from the manuscript and use the free text boxes beneath each contributing author's name in our system to add specific details on the author's contribution. More information is available in our guide to authors.

3/ Figures and Appendix:

- Please make sure that all figures/figure panels are referenced in the manuscript text (currently, a callout is missing for Fig. 6H). Please correct the callout to "Appendix Table S1".
- Please note that you could include Appendix Table S1 as an editable table in the main manuscript text. If you choose to keep an Appendix file, please add a table of content.
- While performing our standard figure check, we noted an anomaly in Fig. 5G & Fig. EV3. Please carefully check the panels composition, correct, and provide a explanation/clarification.

4/ Thank you for providing Source Data. Please upload them as 1 file/figure for the main figures. Source Data for EV figures should be zipped together.

5/ Checklist:

Please fill in the subsection 'inclusion/exclusion criteria' in the section 'Experimental study design and statistics'.

6/ Please provide a synopsis text and image:

Every published paper now includes a 'Synopsis' to further enhance discoverability. Synopses are displayed on the journal webpage and are freely accessible to all readers. They include a short stand first (maximum of 300 characters, including space) as well as 2-5 one-sentences bullet points that summarizes the paper. Please write the bullet points to summarize the key NEW findings. They should be designed to be complementary to the abstract - i.e. not repeat the same text. We encourage inclusion of key acronyms and quantitative information (maximum of 30 words / bullet point). Please use the passive voice. Please attach these in a separate file or send them by email, we will incorporate them accordingly.

Please also suggest a visual abstract to illustrate your article as a PNG file 550 px wide x 300-600 px high. A cropped portion of this image will serve as thumbnail for the table of content on our webpage.

7/ As part of the EMBO Publications transparent editorial process initiative (see our Editorial at

<http://embomolmed.embopress.org/content/2/9/329>), EMBO Molecular Medicine will publish online a Review Process File (RPF) to accompany accepted manuscripts.

This file will be published in conjunction with your paper and will include the anonymous referee reports, your point-by-point response and all pertinent correspondence relating to the manuscript. Let us know whether you agree with the publication of the RPF.

I look forward to receiving your revised manuscript.

Yours sincerely,

Lise Roth

**** Reviewer's comments ****

Referee #1 (Comments on Novelty/Model System for Author):

Appropriate experiments are selected to demonstrate the effects of HPK1 and its loss/inhibition on inflammation-mediated outcomes in experimental stroke in mice. Although effects of HPK1 have been demonstrated in experimental inflammation, the authors here apply that knowledge in clinically relevant experimental models of disease.

Referee #1 (Remarks for Author):

My comments in my review of the first version have been very capably and adequately addressed by the authors. I can see there have been similar attempts to address the comments from the other reviewers likewise. I am satisfied with the responses to my comments.

Referee #2 (Comments on Novelty/Model System for Author):

It's an animal model so the immediate medical impact is always going to be low.

Referee #2 (Remarks for Author):

The authors have made significant improvements to the manuscript, it now reads much better and their use of expanded view figures allows the reader to not get bogged down in the data. I would still argue that figures 3C and 3K (for example) don't really need to be there because there are summary graphs for those data elsewhere in the figures but I will leave this to the discretion of the editor.

My only other comment is regarding the inclusion of female animals. These should just be included with the original figure. So just add all the stroke infarct volume measurements (for example) to the stroke infarct volume measurements for male animals. You increase the overall power of the experiment and you can still just pull out the effect of sex using the correct statistics. This doesn't need to be a full paragraph, you can just put a comment along the lines of 'we used both sexes and found no significant effect of sex on any of the outcomes measured'.

Referee #3 (Remarks for Author):

The authors substantially revised the manuscript.

Responses to the Reviewers' comments

Manuscript number: EMM-2024-20638-V3

Title: Targeting HPK1 inhibits neutrophil responses to mitigate post-stroke lung and cerebral injuries

Authors: Tingting Zhang, Ying Sun, Jing Xia, Hongye Fan, Dingfang Shi, Qian Wu, Ming Huang, Xiao-Yu Hou

Response to Referee #1

(Comments on Novelty/Model System for Author): Appropriate experiments are selected to demonstrate the effects of HPK1 and its loss/inhibition on inflammation-mediated outcomes in experimental stroke in mice. Although effects of HPK1 have been demonstrated in experimental inflammation, the authors here apply that knowledge in clinically relevant experimental models of disease.

(Remarks for Author): My comments in my review of the first version have been very capably and adequately addressed by the authors. I can see there have been similar attempts to address the comments from the other reviewers likewise. I am satisfied with the responses to my comments.

Response: We really appreciate your time and efforts in reviewing our manuscript and providing valuable and insightful suggestions, which help us to provide a better account of this study.

Response to Referee #2

(Comments on Novelty/Model System for Author): It's an animal model so the immediate medical impact is always going to be low.

(Remarks for Author): The authors have made significant improvements to the manuscript, it now reads much better and their use of expanded view figures allows the reader to not get bogged down in the data.

Comment 1: I would still argue that figures 3C and 3K (for example) don't really need to be there because there are summary graphs for those data elsewhere in the figures but I will leave this to the discretion of the editor.

Response: We greatly appreciate your constructive and valuable comments for improving our study. Thank you for this suggestion. We have removed related panels that showing the gating strategies and representative flow cytometry images in Fig 3, Fig 4, Fig 5, Fig 6, Fig 8, and EV figures. Moreover, we have added gating figures into Appendix Figure S1.

Comment 2: My only other comment is regarding the inclusion of female animals. These should just be included with the original figure. So just add all the stroke infarct volume measurements (for example) to the stroke infarct volume measurements for male animals. You increase the overall power of the experiment and you can still just pull out the effect of sex using the correct statistics. This doesn't need to be a full paragraph, you can just put a comment along the lines of 'we used both sexes and found no significant effect of sex on any of the outcomes measured'.

Response: Per your recommendation, we have added the data of outcomes from female mice into Fig 5G-K and Fig 6J-L. Additionally, we have removed the related paragraph in the Results section and put a comment along the lines of “We used both sexes and found no significant effect of sex on post-stroke outcomes measured (**Fig 5G-K** and **Fig 6J-L**).”

Response to Referee #3

(Remarks for Author): The authors substantially revised the manuscript.

Response: We greatly appreciate your time and efforts in reviewing our manuscript and providing valuable and thoughtful suggestions, which help us to provide a better account of this study.

10th Mar 2025

Dear Prof. Hou,

Thank you for submitting your revised files. I am pleased to inform you that your manuscript is accepted for publication and is now being sent to our publisher to be included in the next available issue of EMBO Molecular Medicine.

If you have any questions, please do not hesitate to contact the Editorial Office.

Thank you for your contribution to EMBO Molecular Medicine.

Yours sincerely,

Lise Roth
